# Microbiota-assisted iron uptake promotes immune tolerance in the intestine

Lizhen Zhu[1,2,3,4], Geng Li [1,2,3,4], Zhixin Liang[1,2,3,4], Tuan Qi[1,2,3,4], Kui Deng [1,2,3,4], Jiancheng Yu[1,2,3,4], Yue Peng[1,2,3,4], Jusheng Zheng [1,2,3,4], Yan Song[5] & Xing Chang [1,2,3,4] ✉

Iron deficiencies are the most common nonenteric syndromes observed in patients with inflammatory bowel disease, but little is known about their impacts on immune tolerance. Here we show that homeostasis of regulatory T cells in the intestine was dependent on high cellular iron levels, which were fostered by pentanoate, a short-chain fatty acid produced by intestinal microbiota. Iron deficiencies in Treg caused by the depletion of Transferrin receptor 1, a major iron transporter, result in the abrogation of Treg in the intestine and lethal autoimmune disease. Transferrin receptor 1 is required for differentiation of c-Maf⁺ Treg, major constituents of intestinal Treg. Mechanistically, iron enhances the translation of HIF-2α mRNA, and HIF-2α in turn induces c-Maf expression. Importantly, microbiota-produced pentanoate promotes iron uptake and Treg differentiation in the intestine. This subsequently restores immune tolerance and ameliorated iron deficiencies in mice with colitis. Our results thus reveal an association between nutrient uptake and immune tolerance in the intestine.

As an essential nutrient, iron has a direct impact on the immune system[1,2], and iron deficiencies and iron overload are thus closely related to immune disorders[3,4]. Notably, iron deficiencies are frequently observed in patients with inflammatory bowel diseases (IBDs, including Crohn's disease and ulcerative colitis)[5,6], resulting from increased bleeding into the intestinal lumen, gut microbiota dysbiosis[7], and inflammatory responses[8,9]. Moreover, iron deficiency has been observed in patients prior to the onset of IBD[10], and systemic iron levels have been inversely associated with the risk for IBD in some prospective analyses[5,10]. These clinical observations suggest a causal effect of reduced iron uptake on the progression of IBD, but whether and how iron homeostasis impacts the maintenance of immune tolerance in the intestine have not been elucidated.

In addition to its critical roles in many enzymatic reactions[11–13], iron directly regulates gene expression to enable the rapid response of cells to environmental cues[14–16]. The most well-characterized iron-regulated genes contain iron-responsive elements (IREs), which are conserved short stem–loops found in either the 5' untranslated regions (UTR) or 3'UTRs of genes involved in iron homeostasis (e.g., transferrin receptor, ferritin light chain)[17]. For example, transferrin receptor is harboring five IREs in its 3'UTR. Intracellular iron modulates interactions between IREs and RNA-binding proteins, including iron regulatory proteins (i.e., IRP1 and IRP2), thus enabling dynamic gene expression in response to perturbations of iron homeostasis. When intracellular iron is scarce, IRPs bind to IREs and block the translation of genes that are involved in the uptake and storage of iron, such as Ferritin and Transferrin receptor, which helps to conserve iron and prevent it from being stored in excess[18]. Conversely, when iron levels are high, IRPs dissociate from IREs, allowing the translation of the Ferritin and Transferrin receptor and enabling cells to take up and store excess iron. Although initially identified in genes participating in iron homeostasis, IREs are also found in genes not directly involved in

¹Key Laboratory of Growth Regulation and Translational Research of Zhejiang Province, School of Life Sciences, Westlake University, Hangzhou, Zhejiang, China. ²Center for Infectious Disease Research, Westlake Laboratory of Life Sciences and Biomedicine, Hangzhou, Zhejiang, China. ³Research Center for Industries of the Future (RCIF), Westlake University, Hangzhou, Zhejiang, China. ⁴Institute of Basic Medical Sciences, Westlake Institute for Advanced Study, Hangzhou, Zhejiang, China. ⁵School of Medicine, University of California San Diego, La Jolla, CA, US. ✉e-mail: changxing@westlake.edu.cn

iron homeostasis. The importance of iron-mediated gene regulation for processes other than the maintenance of iron homeostasis remains unknown.

Regulatory T cells (Tregs) are a specialized lineage of helper T cells expressing the lineage transcription factor Foxp3[19] and are essential for the maintenance of self-tolerance, including immune tolerance in the intestine[20,21]. Similar to conventional T cells, regulatory T cells further differentiate into effector cells (effector Tregs or eTregs) to fully execute their suppressive functions[22], a process often accompanied by the acquisition of other transcription factors (e.g., T-bet, Bcl6, Gata-3, RORγt)[23]. The variegated expression of these transcription factors further specifies the transcriptomes, localization, and effector functions of Tregs, giving rise to diverse effector Treg subsets residing in lymphoid and nonlymphoid organs to maintain tissue homeostasis and tolerance[23]. As a prototypical Treg subset in non-lymphoid organs, Tregs residing in the lamina propria of the intestine require continuous interaction with the intestinal microbiota to sustain the expression of specialized transcription factors (e.g., RORγt)[24–27] and acquire gut-specific phenotypes and metabolic patterns to maintain intestinal homeostasis[28–30]. The mechanisms by which the microbiome regulates intestinal Treg homeostasis have not been fully elucidated.

In this study, we showed that the differentiation of regulatory T cells in the intestinal mucosa was critically dependent on their higher levels of intracellular iron, the uptake of which was promoted by pentanoate, a microbiota-produced short-chain fatty acid (SCFA). Decreased cellular iron levels in regulatory T cells via the specific depletion of Transferrin receptor I (TfR1) resulted in lower cellular iron levels and early-onset lethal autoimmune disease. Under noninflammatory conditions or upon acute inactivation, TfR1 deficiency specifically abolished Treg cells in the intestine, leading to intestinal inflammation. Mechanistically, intracellular iron promoted the expression of hypoxia-induced factor 2α (HIF-2α) via IREs located in its 5′-UTR, and the elevated HIF-2α level further enhanced the expression of c-Maf, a transcription factor critical for the differentiation of intestinal Treg cells. More importantly, microbiota-produced pentanoate promoted iron procurement and c-Maf⁺ Treg differentiation in the intestine, alleviating both colonic inflammation and iron deficiency in a murine model of colitis. Together, our data revealed a previously unrecognized association between iron metabolism and Treg homeostasis in the intestine, underscoring the importance of commensal bacteria in promoting immune tolerance by modulating nutrient procurement.

## Results

### TfR1-mediated iron uptake by Treg cells is critical for the maintenance of immune tolerance

To analyze the impacts of iron deficiency on regulatory T cells and immune tolerance, we specifically depleted TfR1 (coded by the *Tfrc* gene), a major iron transporter, in Treg cells by crossing *Tfrc^{loxp/loxp}* mice[16] with *Foxp3^{YFP-IRES-Cre}* mice[31]. In the resulting *Tfrc^{fl/fl} Foxp3^{YFP-IRES-Cre/y}* mice (referred to as *Tfrc* cKO mice thereafter), TfR1 was specifically deleted in Treg cells (Fig. 1a, Supplementary Fig. S1a), resulting in total intracellular iron levels reduced by ~50% (8.58 ng/10⁶ cells vs 4.10 ng/10⁶ cells) as determined by inductively coupled plasma−mass spectrometry (ICP−MS) (Fig. 1b). The partial reduction in the cellular iron concentration in TfR1-deficient Treg cells was in line with previous studies showing that T cells utilize nontransferrin-bound iron (NTBI)[32,33], the uptake of which is independent of TfR1.

Profoundly, all *Tfrc* cKO mice died from the autoinflammatory disease within 4 weeks of age (Fig. 1c). As early as three weeks after birth, *Tfrc* cKO mice exhibited signs of systemic inflammation (smaller body size, dermatitis, and squinty eyes), similar to Foxp3-deficient mice (e.g., Scurfy mice)[34,35] (Supplementary Fig. S1b). Further analysis revealed an enlarged spleen and lymph nodes (Supplementary Fig. S1c)

as well as massive leukocyte infiltration into the skin, liver, intestine, and lungs (Fig. 1d) in *Tfrc* cKO mice. The infiltrating leukocytes in the peripheral organs were mainly T cells and myeloid cells (Supplementary Fig. S1d), while B cells were depleted in the spleen, PBMC, bone marrow, and liver of the *Tfrc* cKO mice. The accumulation of T cells in the periphery was mainly caused by increased cellular proliferation as determined by Ki-67 staining (Supplementary Fig. S2a) rather than diminished cell death (Supplementary Fig. S2b). The thymus displayed severe atrophy in the 3-week-old *Tfrc* cKO mice (Supplementary Fig. S1e). Furthermore, the numbers of effector/memory population (CD44^{hi}CD62L^{lo}) of T cells (Fig. 1e, Supplementary Fig. S1f) were drastically increased in the 3-week-old *Tfrc* cKO mice compared to their wild-type (WT) littermates. Additionally, flow cytometric analysis revealed that the numbers of splenic CD4 T cells secreting IL-17A, IFN-γ or IL-4 were significantly increased in *Tfrc* cKO mice (Fig. 1f, ` Fig. S2c), and CD8 T cells in *Tfrc* cKO mice had elevated IFN-γ levels (Supplementary Fig. S2d). Collectively, these data demonstrate that loss of TfR1 in Treg cells results in fatal autoimmune inflammatory responses.

Despite the importance of iron for many important cellular processes, loss of TfR1 does not impair the development of most nonerythroid tissues, including immune cells[16,36,37]. Similarly, *TfR1* deficiency only moderately diminished the homeostasis of Treg cells in the periphery. In 6-day-old *Tfrc* cKO mice that had not yet displayed notable T-cell activation (Supplementary Fig S3a), the proportions of Treg cells in the spleen were reduced by ~30% (from 10.6% to 7.0%) (Fig. 1g, h). Such a reduction was not caused by Treg developmental defects because the Treg numbers in the thymus were comparable between *Tfrc* cKO and WT mice (Supplementary Fig. S3b, c). The proportion of Tregs in the spleen was further reduced in 3-week-old *Tfrc* cKO mice, presumably due to the progression of autoimmune disease and expansion of activated conventional T cells (Fig. 1g, h).

Although TfR1 may function beyond taking up iron-bound transferrin[38], the lethal autoimmunity in *Tfrc* cKO mice was caused solely by selective iron deficiencies. After treatment with iron dextran, an iron source that is taken up independently of TfR1, *Tfrc* cKO mice were rescued from early lethality and survived for up to 200 days (Fig. 1i). In the iron dextran-treated *Tfrc* cKO mice, the percentages of Treg cells (Supplementary Fig. S3f) and activated/memory T cells (Supplementary Fig. S3g) were restored to levels similar to those in the WT littermates. Together, these data indicate that TfR1-mediated iron uptake is indispensable for the maintenance of immune tolerance by regulatory T cells in vivo.

### Iron promotes the differentiation of c-Maf⁺ eTreg cells

Because *Tfrc* cKO mice developed the lethal autoimmune disease by three weeks of age, we analyzed heterozygous female *Tfrc^{fl/fl} Foxp3^{YFP-Cre/+}* mice to circumvent the possible secondary effects of T-cell activation. In these mice, because of random X-chromosome inactivation, the Tregs were comprised of a mixture of WT (Foxp3⁺Cre-YFP⁻) and *Tfrc*-deficient (Foxp3⁺Cre-YFP⁺) cells. In contrast to *Tfrc* cKO (*Tfrc^{fl/fl} Foxp3^{YFP-IRES-Cre/y}* mice) mice, adult *Tfrc^{fl/fl} Foxp3^{YFP-Cre/+}* mice displayed no detectable inflammatory phenotypes and had normal percentages of activated/memory T cells in the spleen (Fig. S3h). In these mice, the ratios of Foxp3⁺YFP⁺ cells among CD4 T cells were reduced by ~30% compared to those of their counterparts in *Tfrc^{+/+} Foxp3^{YFP-Cre/+}* mice (from 3.04% to 2.08%) (Fig. 2 a, b), a magnitude similar to that of the Treg reduction in 6-day-old *Tfrc* cKO mice (Fig. 1g). Despite the partial loss of Treg cells, Foxp3 expression on a per-cell basis was comparable between TfR1-deficient and WT Treg cells (Fig. 2c).

To investigate which subsets of Treg cells were strictly dependent on TfR1-mediated iron uptake, we conducted single-cell RNA sequencing (scRNA-seq) analysis of total Treg cells (CD25⁺CD4⁺) isolated from heterozygous *Tfrc^{fl/fl} Foxp3^{YFP-Cre/+}* mice (Fig. 2d). YFP⁻ Treg cells (Cre-null and Tfrc-undeleted, *n* = 6307) and YFP⁺ Tregs (Cre-positive and Tfrc-deleted, *n* = 247) were subsequently separated in silico based on

the expression of the YFP-IRES-Cre transgene. Consistent with previous studies[39,40], Treg cells in the spleen were mainly separated into two populations, central Tregs (cTregs, cluster B) and effector Tregs (eTregs, cluster A). Compared to the Cre⁻ Treg cells (*aka* WT), the Cre⁺ Treg cells (*aka Tfrc*-deficient cells) had diminished frequencies of the eTreg population (Fig. 2f). We further analyzed the top differentially

expressed genes between WT and *Tfrc*-deficient Tregs in our scRNA-seq datasets. Notably, the expression of c-Maf, a transcription factor critical for IL-10 production[41] and RORγt⁺ Treg differentiation[29,30], was downregulated most significantly in *Tfrc*-deficient Treg cells compared to their WT counterparts (Fig. 2f, g). These data suggest that intracellular iron is critical for the differentiation of c-Maf⁺ Treg cells.

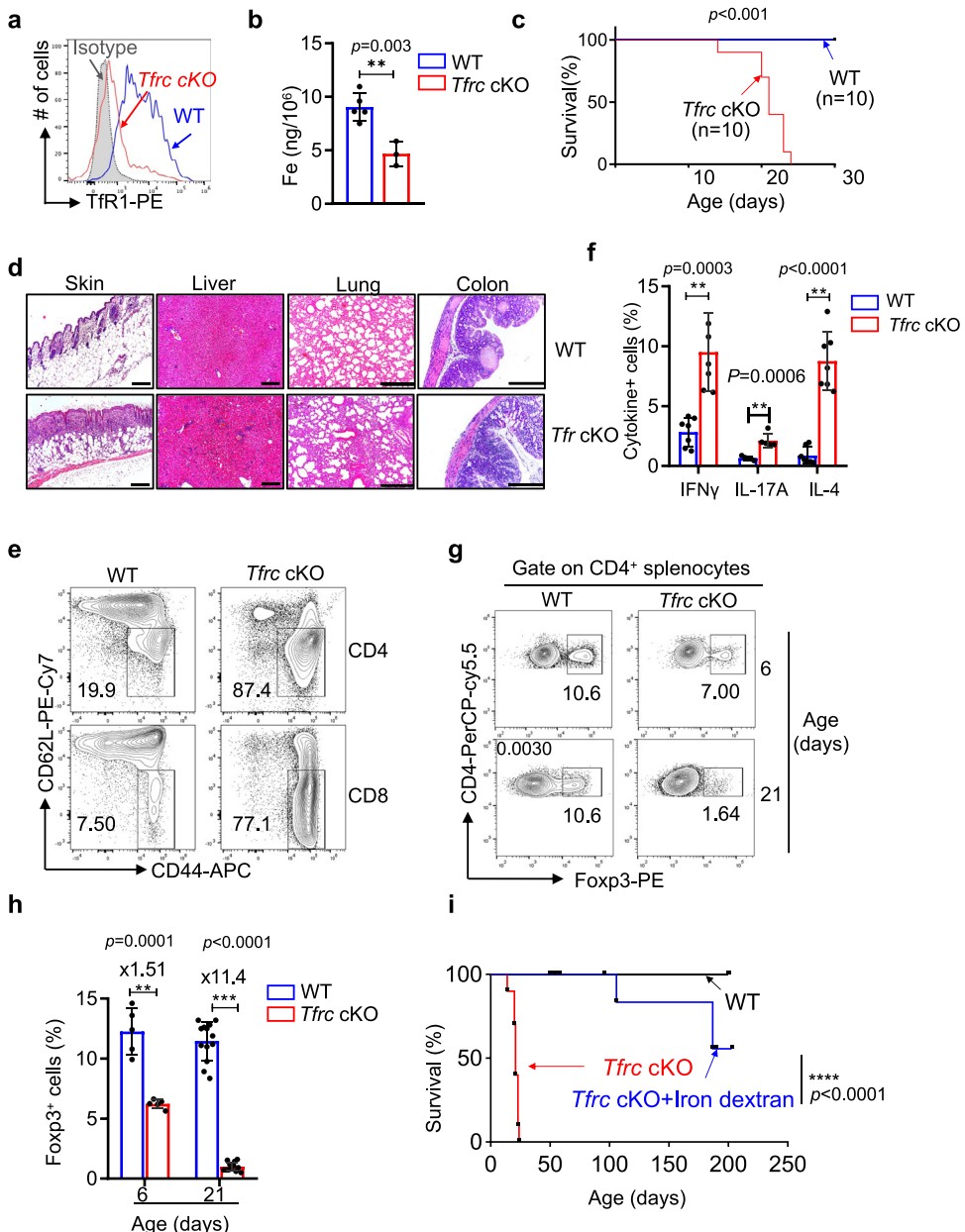

**Fig. 1 | Loss of TfR1 in Treg cells results in lethal autoimmunity. a** Representative histogram comparing the expression of TfR1 in splenic Treg cells (CD4⁺Foxp3⁺) from 3-week-old *Tfrc* cKO mice (*Tfrc*^fl/fl *Foxp3*^YFP-IRES-Cre/y mice) and their WT littermates (*Tfrc*^+/+ *Foxp3*^YFP-IRES-Cre/y mice). Grey-shaded areas represent staining with an isotype control antibody. The data are representative of 5 independent experiments. **b** The intracellular iron levels in CD25⁺YFP⁺ cells isolated from either female *Tfrc*^fl/fl *Foxp3*^Cre-YFP/+ (*Tfrc* cKO Treg) or *Tfrc*^+/+*Foxp3*^Cre-YFP/+ (WT Treg) mice were determined by ICP−MS(*n* = 5 and 3 mice from 3 independent experiments for WT and *Tfrc* cKO respectively) . **c** Kaplan−Meier survival curves of *Tfrc* cKO mice and WT littermates. **P < 0.001 based on Mantel−Cox test. **d** Representative H&E staining of the indicated tissues of 3-week-old *Tfrc* cKO mice and their WT littermates. The data are representative of at least three independent experiments. Scale bars represent 100 μm (skin and liver) or 200 μm (lung and colon). **e** CD4 (upper

panels) and CD8 (lower panels) splenocytes were examined for the expression of CD44 and CD62L, and the percentages of the effector/memory population (CD44^hiCD62L^lo) are indicated. The data are representative of eight independent experiments. **f** A summary of the percentages of IFNγ-, IL-17A- or IL-4-secreting CD4 T cells from 21-day-old WT or *Tfrc* cKO mice. (*n* = 7 mice for IFNγ; 5 mice for IL-17A; 7 mice for IL-4 from 5 independent experiments). **g**, **h** CD4⁺ splenic T cells of *Tfrc* cKO and WT mice were analyzed for intracellular Foxp3 expression at either 6 days (upper panels) or 21 days (lower panels) of age. The data are a representative (**g**) or summary (**h**) of at least five independent experiments (*n* = 5 for 6-day-old mice,; *n* = 13 for 21-day-old mice).(**b**, **f**, **g**). **P < 0.01, ***P < 0.001 in two-tailed Student's *t* test. Data shown are mean ± SD. **i** Neonatal *Tfrc* cKO mice were treated with iron dextran. Kaplan−Meier survival curves of untreated *Tfrc* cKO, WT mice and *Tfrc* cKO mice receiving iron dextran treatment. ****P < 0.0001 based on Mantel−Cox test.

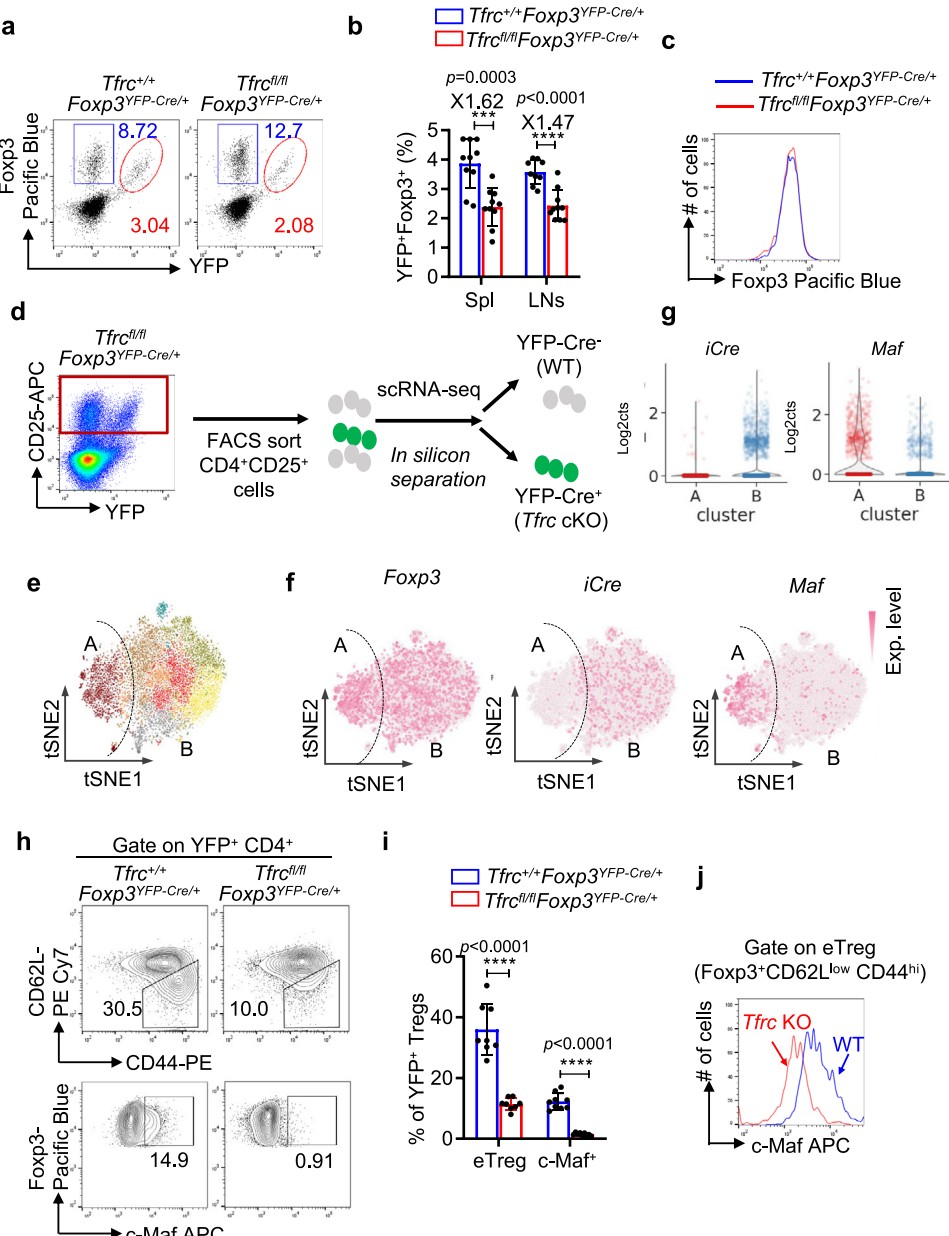

**Fig. 2 | TfR1 promotes the differentiation of c-Maf⁺ Treg cells. a–c** CD4⁺ splenic cells from *Tfrc*^fl/fl *Foxp3*^YFP-Cre/+ and *Tfrc*^+/+*Foxp3*^YFP-Cre/+ (WT) female mice were gated and analyzed for YFP and Foxp3 expression. **a** Representative flow cytometric analysis with pregating on CD4⁺ cells; (**b**). Summary of the frequencies of YFP⁺ Foxp3⁺ cells in CD4⁺ T cells from the spleen (*n* = 10 mice) and lymph nodes (*n* = 10 for *Tfrc*^fl/fl *Foxp3*^YFP-Cre/+ mice and *n* = 9 for WT mice); (**c**). Histogram comparing Foxp3 levels in gated CD4⁺Foxp3⁺YFP⁺ cells from the spleen of either *Tfrc*^fl/fl *Foxp3*^YFP-Cre/+ or *Tfrc*^+/+*Foxp3*^YFP-Cre/+ female mice. The data are summary (**b**) or representative (**a, c**) of five independent experiments. **d–g** scRNA-seq analysis of CD4⁺CD25⁺ cells isolated from the spleens of *Tfrc*^fl/fl *Foxp3*^YFP-Cre/+ mice. TfR1-deficient cells (YFP⁺) and WT (YFP⁻) cells were separated in silico based on detection of the YFP-IRES-CRE transcript. **d** Schematic of the experimental design. **e** t-SNE projection of all CD4⁺CD25⁺ cells isolated from *Tfrc*^fl/fl *Foxp3*^YFP-Cre/+ mice. YFP⁺ (Cluster A) and YFP⁻ (Cluster B) cells are indicated; **f**, t-SNE projection of all cells color-coded by normalized counts for *Foxp3* (left), *Cre-YFP* (middle), or *Maf* (right); **g**, Violin plots comparing Cre-YFP (left) and *Maf* (right) expression between cluster A and cluster B. **h, i** As in (**a**), CD4⁺ FOXP3⁺YFP⁺ cells from the spleens of *Tfrc*^fl/fl *Foxp3*^YFP-Cre/+ mice were analyzed for the expression of CD62L and CD44 (upper panels) or c-Maf (lower panels). The frequencies of effector Treg cells (CD62L^low CD44^hi, upper panels) and c-Maf⁺ Tregs (lower panels) are indicated. (**h** Representative flow cytometric staining; (**i**). Summary of the c-Maf⁺ Treg cell and eTreg cell percentages (*n* = 8 mice for eTreg analysis; *n* = 9 mice for c-Maf⁺ Treg analysis based on 3 independent experiments). YFP⁺ cells from the *Tfrc*^+/+*Foxp3*^YFP-Cre/+ mice were included as controls. **j** As in (**a**), effector Treg cells in the spleen were gated as CD44^hiCD62L^low Foxp3⁺YFP⁺, and c-Maf expression was compared between TfR1-deficient and WT eTreg cells by flow cytometry. The data are representative of three independent experiments. **b–i** Data shown are mean ± SD, ***p < 0.001, ****p < 0.0001 by two-tailed Student's *t* test.

The reduction in the percentage of c-Maf⁺ cells among TfR1-deficient Treg cells was further validated by flow cytometry. In female *Tfrc*^+/+*Foxp3*^YFP-Cre/+ mice, eTregs (CD62L^low CD44^hi) and c-Maf⁺ cells constituted ~30% and ~15% of YFP⁺ Tregs, respectively, while their ratios among YFP⁺ cells (*aka* TfR1-deficient Tregs) in the *Tfrc*^fl/fl

*Foxp3*^YFP-Cre/+ mice were reduced to less than 10 and 1%, respectively (Fig. 2h, i). A similar reduction in c-Maf⁺ Treg cell numbers was observed in 6-day-old *Tfrc* cKO mice (Supplementary Fig. S3d, e). Moreover, on a per-cell basis, TfR1-deficient eTreg cells had reduced levels of c-Maf but not Foxp3 compared to those of their WT

counterparts (Fig. 2j), indicating that iron was critical for the expression of c-Maf in Treg cells.

The cell-autonomous function of TfR1 in regulating c-Maf+ Treg cell differentiation was further confirmed in mixed bone marrow chimeric mice. We constructed bone marrow chimeric mice by transferring *Tfrc* cKO bone marrow cells (CD45.2+) mixed at a 1:1 ratio with congenic WT bone marrow cells (CD45.1+) into *Rag1-/-* mice (Supplementary Fig. S4a). The presence of WT Treg cells in these mice (CD45.1+) prevented T-cell activation and autoimmune disease development (Supplementary Fig. S4b). Similar to the TfR1-deficient cells in the heterozygous female mice, the ratios of Foxp3+ cells to CD45.2+CD4+ T cells in the spleen were reduced by ~50% compared with those of their WT counterparts (Supplementary Fig. S4c, d). Importantly, few eTreg or c-Maf-expressing Treg cells were observed in TfR1-deficient bone marrow-derived cells (Supplementary Fig. S4e, f). Together, these data demonstrated a cell-intrinsic function of *TfR1* in promoting the differentiation and/or maintenance of a subset of Treg cells expressing c-Maf.

Conversely, elevated intracellular iron levels promoted c-Maf expression in Treg cells. We treated WT mice with iron dextran, which increased the Treg ratio (from 13.3% to 17.4%, Suppl. Fig. S5a, b) and, more importantly, resulted in an ~30% increase in c-Maf+ Treg cells in the spleen (from 26.5 to 39%, Fig. 3a, b). Moreover, the stimulation of in vitro differentiated Tregs (iTregs) with iron sulfate had little effect on the induction of Foxp3 (Supplementary Fig. S4 c, d), but c-Maf expression was substantially enhanced in a dose-dependent manner (Fig. 3c, d). This induction of c-Maf by iron was not caused by increased cellular proliferation because the iron-treated cells undergoing the same numbers of division had higher c-Maf expression than untreated cells (Supplementary Fig. S5 e, f). Finally, depletion of iron with an iron chelator (Desferoxamine, DFO) repressed c-Maf expression in iTreg cells in vitro (Fig. 3c). These data demonstrate that iron directly acts on Treg cells to promote c-Maf expression.

### TfR1-mediated iron uptake is critical for the effector functions of Treg cells

To gain further insight into the TfR1-mediated transcriptional program in Treg cells, we conducted RNA-seq analysis of TfR1-deficient Treg cells (YFP+CD25+CD4+) isolated from female *Tfrc*fl/fl *Foxp3*YFP-Cre/+ mice. Differential expression analysis revealed 219 upregulated genes and 312 downregulated genes in TfR1-deficient Treg cells compared with their counterparts from *Tfrc*+/+ *Foxp3*YFP-Cre/+ mice (Supplementary Fig. S6a). Gene ontology analysis of the differentially expressed genes (DEGs) revealed that "T-cell activation" and "regulation of immune effector processes" were the most dominant biological processes affected by *Tfrc* deficiency (Supplementary Fig. S6b)[23]. Indeed, among the most suppressed genes were key effector molecules of Tregs, including CTLA-4, PD1, ICOS, IL-10, Fgl2, CD103 and Ahr, which were further confirmed by flow cytometric analysis (Supplementary Fig. S6c). Corroborating the importance of TfR1 in effector Treg cells, TfR1 was expressed at higher levels in effector Treg cells than in natural Treg or conventional T cells (Supplementary Fig. S6 d, e).

Consistent with the decreased expression of effector molecules, TfR1-deficient Treg cells lost their suppressive activities in vivo. We isolated TfR1-deficient Treg cells (CD4+YFP+) from 6-day-old *Tfrc*fl/fl *Foxp3*YFP-IRES-Cre/y mice and cotransferred the cells with WT naïve CD4 T cells (CD4+CD45RBhiCD25-) into *Rag1-/-* mice. While the cotransfer of WT Tregs prevented weight loss and intestinal inflammation, mice that received TfR1-deficient Tregs lost weight (Supplementary Fig. S6f) and exhibited severe intestinal tissue damage (Supplementary Fig. S6g), indicating that TfR1-deficient cells succumbed to Treg cells with fewer or no suppressive functions. These results indicate that the acquisition of effector functions by Treg cells is dependent on the level of TfR1-mediated iron uptake.

### Iron induces c-Maf expression by enhancing IRE-dependent HIF-2α expression

To investigate the mechanism by which iron promoted c-Maf expression, we performed unbiased transcriptome analyses of iron-stimulated iTreg cells and untreated iTreg cells. Differential expression analysis revealed 655 upregulated genes and 490 downregulated genes in iTreg cells stimulated with iron (Fig. 3e). Gene ontology analysis of the DEGs revealed that the iron-regulated genes of Treg cells were significantly enriched for cellular responses to decreased oxygen levels (hypoxia) as well as for cellular responses to stress, leukocyte activation and cholesterol biosynthesis processes (Fig. 3f). Gene set enrichment analysis (GSEA)[42,43] of TfR1-deficient Treg cells also showed that the levels of genes associated with the hypoxia response were downregulated compared with those in WT Treg cells (Supplementary Fig. S7a), indicating that partial iron loss might have impaired the expression of hypoxia-responsive genes in Treg cells.

The transcriptional program regulating hypoxia responses is modulated by transcriptional complexes composed of an oxygen-sensitive α-subunit (HIF-1α, HIF-2α, and HIF-3α) and a conserved β-subunit (HIF-1β, also known as aryl hydrocarbon receptor nuclear translocator, AHRN)[44]. Notably, the expression of HIF-2α but not HIF-1α was decreased in TfR1-deficient Treg cells (YFP+CD25+) from the spleens of female *Tfrc*fl/fl *Foxp3*YFP-Cre/+ mice (Fig. 3g). Despite the reduced HIF-2α protein level, HIF-2α mRNA expression was not significantly decreased in TfR1-deficient Treg cells (Supplementary Fig. S7b). Corroborating these findings in TfR1-deficient Treg cells, iron treatment enhanced the protein expression of HIF-2α but not that of HIF-1α in iTreg cells (Fig. 3h), while the mRNA level of HIF-2α was not increased (Suppl. Fig. S7c). Given the presence of IREs in the 5'UTR of HIF-2α[45], the activity of the HIF-2α 5'-UTR in iTreg cells was substantially enhanced by iron stimulation (Supplementary Fig. S7d), while the mRNA levels of the luciferase reporter were not altered. Two mutations that abrogated IRE/IRP interactions in the HIF-2α 5'UTR[45,46] also abolished the stimulatory effects of iron (Supplementary Fig. S7e). These data indicate that the translation of HIF-2α is modulated by intracellular iron via IREs in its 5-UTR[45,46].

More importantly, HIF-2α was shown to be required for the iron-mediated induction of c-Maf expression in Treg cells. HIF-2α was expressed at higher levels in c-Maf hi Treg cells in vivo than in c-Maf low Treg cells (Supplementary Fig. S7f). When iTreg cells were treated with PT2385, which inhibits HIF-2α transcriptional activity[47], the ability of iron to induce c-Maf expression was completely abrogated (Fig. 3i, j). Additionally, knockdown of HIF-2α via a retrovirally transduced shRNA blocked the induction of c-Maf expression by iron (Supplementary Fig. S7 g, h, i). Finally, reanalysis of previously published ChIP-Seq results[48] revealed direct associations of HIF-2α and HIF-1β (AHRN) with the c-Maf promoter region (Supplementary Fig. S7j), suggesting that c-Maf is a direct target of HIF-2α. Collectively, these data demonstrate that the expression of c-Maf in Treg cells is regulated by HIF-2α, the translation of which is promoted by intracellular iron via IREs.

### Accumulation of intestinal Treg cells requires TfR1-mediated iron uptake

c-Maf-expressing Treg cells are predominantly found in the lamina propria of the intestine, express RORγt, and play pivotal roles in maintaining immune tolerance in the intestine[28–30]. Consistent with findings of iron-promoting c-Maf expression in Treg cells, TfR1-mediated iron uptake was shown to be crucial for the homeostasis of Treg cells in the intestine. First, Treg cells in the intestine (including the colon and small intestine) expressed higher levels of TfR1 and contained more labile iron than their counterparts in the spleen and lung (Supplementary Fig. S8a). HIF-2α was also expressed at higher levels in the colonic Tregs than Tregs in the spleen and lung. Importantly, despite having only a minor impact on the number of splenic Treg cells, reduced cellular iron levels led to a complete loss of Treg cells in

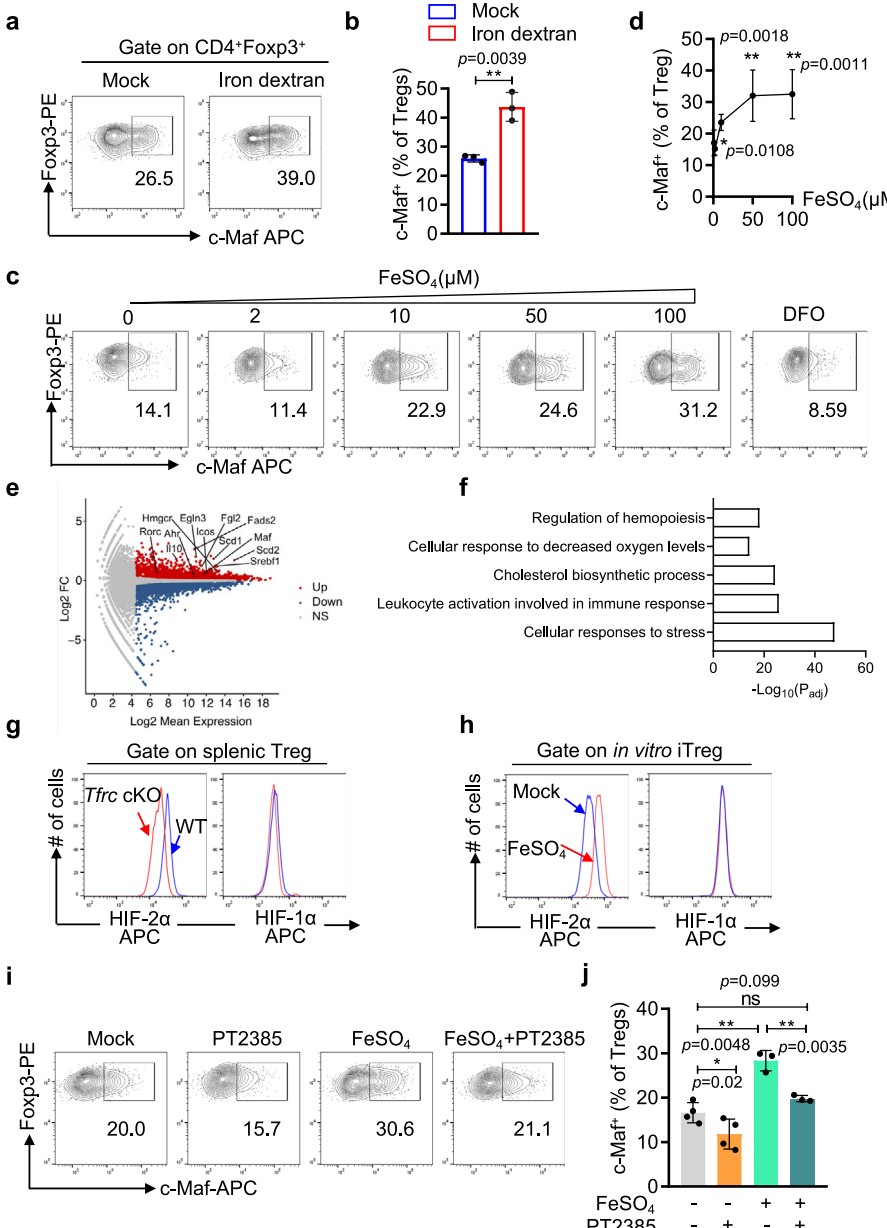

**Fig. 3 | Iron promotes c-Maf expression in Treg cells by modulating HIF-2α expression. a, b** WT mice were administered iron dextran (37.5 mg/e.a) every five days for 3 weeks. The percentages of c-Maf⁺ Treg cells in the spleen were determined by flow cytometry. **a** Representative flow cytometric plots; (**b**). Summary of the percentages of c-Maf⁺ cells in splenic Treg cells from three independent experiments. **c, d** CD4 T cells were activated under iTreg differentiation conditions (anti-CD3+anti-CD28 + TGFβ1 + IL-2+anti-IFNγ+anti-IL4) with the indicated concentrations of ferrous sulfate or DFO. The expression of c-Maf and Foxp3 was determined by intracellular staining four days after T-cell activation. **c** Representative flow cytometry analysis; (**d**). Summary of the percentages of c-Maf⁺ Tregs ($n = 7$ biologically independent experiments for mock, and $n = 5$ for FeSO₄-treated mice). (**e, f**). As in (**c**), the transcriptomes of ferrous sulfate-treated iTreg cells were compared to those of iTreg cells without treatment by RNA-seq. **e** M-A plots showing the differentially expressed genes; Differentially expressed genes were determined by DESeq2 and are shown in red or blue ($|LFC| \geq 0.5$,

$p < 0.05$ in Wald test). **f** GO analysis of the differentially expressed genes using Metascape. $p$-values are calculated based on the cumulative hypergeometric distribution, and q-values are calculated using the Benjamini-Hochberg procedure. **g** The expression of HIF-1α and HIF-2α in YFP⁺Foxp3⁺CD4⁺ cells of *Tfr1*^fl/fl *Foxp3-Cre*^YFP/+ mice (*Tfrc* cKO) was determined by flow cytometry and compared with that in their counterparts (WT) of *Tfrc*^+/+*Foxp3-Cre*^YFP/+ mice. The data are representative of three independent experiments. **h** As in (**c**), one day after T-cell activation, iTregs were cultured with 50 µM FeSO₄ for 24 h, and the expression of HIF-2α and HIF-1α in Foxp3⁺ cells was determined by intracellular staining. The data are representative of three independent experiments. **i, j** As in (**a**), iTreg cells were treated with 50 µM ferrous sulfate with or without 50 µM PT2385 (an HIF-2α inhibitor). The expression of c-Maf was determined by intracellular staining. **i** Representative results. **j** Summary of c-Maf⁺% cells in iTreg cells based on four independent experiments. **b, d, j** Data shown are mean ± SD, *$p < 0.05$, **$p < 0.01$,***$p < 0.001$ by two-tailed Student's t test.

the intestine. Analysis of the female *Tfrc*^fl/fl *Foxp3*^YFP-Cre/+ mice revealed few YFP⁺Foxp3⁺ (*i.e.*, TfR1 KO Treg) cells in the lamina propria of the small intestine (S.I.) or the colon, while their ratios in the lung and spleen were reduced by only ~30% when compared to those of their WT counterparts (YFP⁺Foxp3⁺ cells from *Tfrc*^+/+ *Foxp3*^YFP-Cre/+ mice)

(Fig. 4a, b). Such a selective absence of TfR1-deficient Treg cells in the intestine was also observed in the mixed bone marrow chimeric mice, as *Tfrc* cKO bone marrow-derived Treg cells (CD4⁺CD45.2⁺Foxp3⁺) were nearly absent in the small intestine and colon (Supplementary Fig S4c, d). These data collectively indicate that Treg cells in the lamina

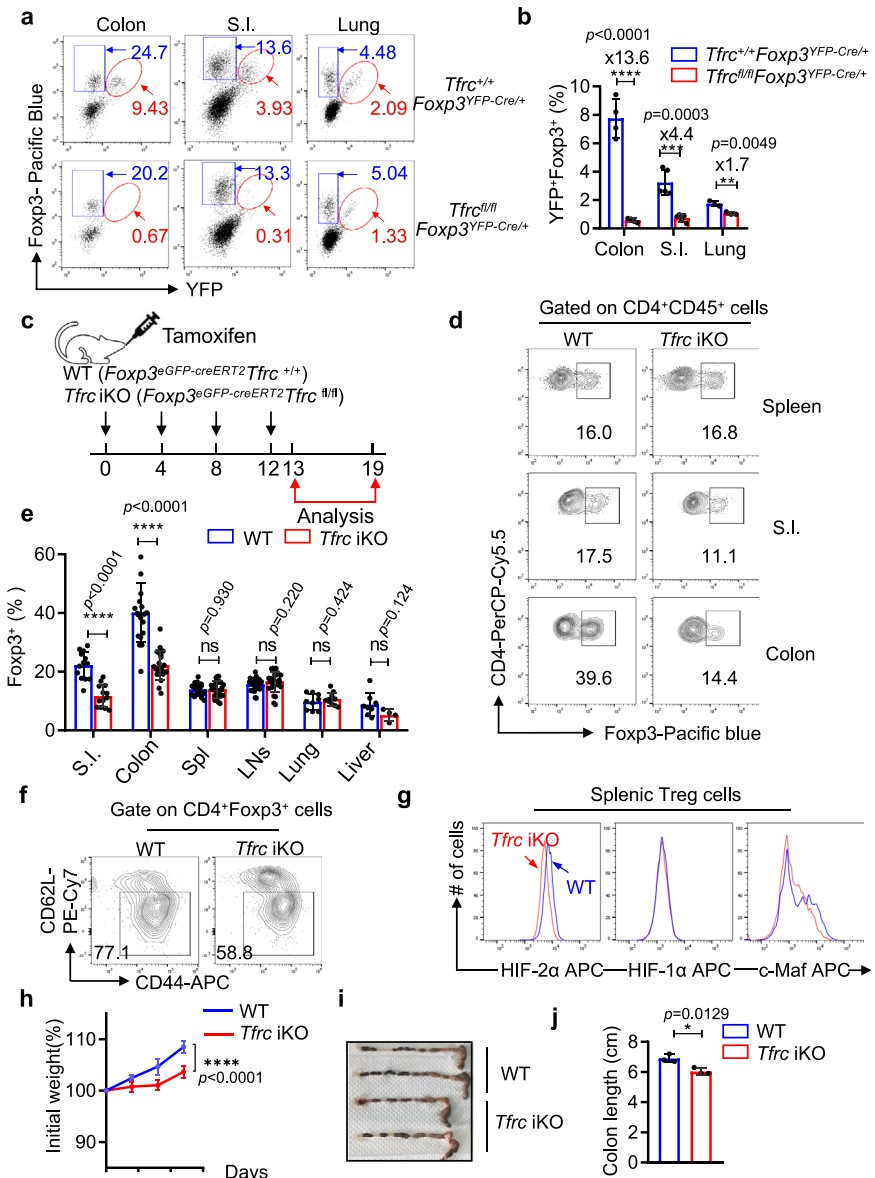

**Fig. 4 | Continuous iron uptake is required for the homeostasis of intestinal Treg cells. a, b** Six- to eight-week-old female *Tfrc*fl/fl *Foxp3*YFP-Cre/+ and *Tfrc*+/+*Foxp3*YFP-Cre/+ mice were analyzed for tissue-resident Treg cells. CD4+ T cells from the lung, lamina propria of the large intestine (L.I.) and small intestine (S.I.) were isolated and analyzed for YFP and Foxp3 expression by intracellular staining. **a** Representative flow cytometry analysis showing the frequencies of YFP+Foxp3+ (red gates) and YFP-Foxp3+ (blue gates) cells in the CD4+CD45+ population; **b** Summary of the percentages of YFP+Foxp3+ cells in total CD4 T cells from the indicated tissues. (colon, *n* = 4; S.I., *n* = 5; lung, *n* = 3 mice examined over three independent experiments). **c** Schematic of acute TfR1 inactivation in Treg cells. **d, e** Fourteen days after the initial tamoxifen treatment, the ratios of Treg cells to CD4 T cells in the indicated tissues were analyzed by the intracellular staining of Foxp3. **d** Representative flow cytometry analysis. **e** Summary of the proportions of Foxp3+ CD4+CD45+ T cells from the indicated tissues. The data are representative of (**d**) or summary (**e**) of at least three independent experiments.(S.I., *n* = 13; colon, *n* = 21; spleen, *n* = 27; LNs,

*n* = 27; lung, *n* = 9, *n* = 10 for WT or *Tfrc* iKO mice respectively; liver, *n* = 8, *n* = 7 for WT or *Tfrc* iKO mice respectively). **f** Treg cells in the spleen were gated as Foxp3+CD4+, and the proportion of effector Treg cells (CD44hiCD62Llow) was determined. The data are representative of three independent experiments. **g** The levels of the indicated transcription factors in CD4+Foxp3+ cells from the spleen were determined by intracellular staining. The data are representative of three independent experiments. **h–j** Acute inactivation of TfR1 resulted in colitis, manifested as slower weight gain and colon shortening, in tamoxifen-treated *Foxp3*eGFP-creERT2*Tfrc* fl/fl mice. **h** Summary of weight changes following tamoxifen treatment (*n* = 9 mice per group examined over 2 independent experiments)****p < 0.0001 by a two-tailed *t test*. Data are mean ± SEM; **i** Representative photos showing the colon lengths. **j** Summary of the colon lengths (*n* = 3 mice from 2 independent experiments).**b, e, j** Data shown are mean ± SD, *p < 0.05, **p < 0.01,***p < 0.001, ****p < 0.0001 by two-tailed Student's *t* test.

propria of the intestine stringently rely on high levels of intracellular iron for their differentiation and homeostasis.

Furthermore, only intestinal Treg cells and not Treg cells residing in other tissues continuously required TfR1-mediated iron uptake. We acutely deleted *Tfrc* in mature Treg cells by injecting tamoxifen into adult *Foxp3*eGFP-creERT2*Tfrc*fl/fl mice (referred to as *Tfrc* iKO hereafter). In

these mice, tamoxifen activated an ERT2-Cre fusion protein under the control of the *Foxp3* promoter, enabling the specific and acute abrogation of TfR1 in Treg cells (Fig. 4c, Suppl. Fig. S8b). Consequently, levels of intracellular iron in Treg cells were reduced following tamoxifen treatment (Supplementary Fig. S8b). Three weeks after the initial tamoxifen treatment, the ratios of T cells (Supplementary

Fig. S8c, d) and Treg cells (Fig. 4d, e) were not altered in the spleens of *Tfrc* iKO mice, while eTreg populations (Fig. 4f) and the expression of c-Maf and HIF-2α were diminished, confirming that the expression of these transcription factors was actively modulated by the iron levels in Treg cells (Fig. 4g), which was essential for eTreg cells development. In contrast to those in the spleen, the proportions of Treg cells in the colon and small intestine (S.I) were reduced by more 50% (colon from 39.6% to 14.4%; S.I. 17.5% to 11.1%) following tamoxifen treatment (Fig. 4d, e). Such reductions were unique for intestinal Treg cells because the ratios of Treg were not changed in other nonlymphoid organs, including the lung and liver (Fig. 4e). Given that Treg cells in the lamina propria of the intestine had a decay and exchange rate over 13 weeks[49], the diminished Treg cell numbers three weeks after tamoxifen treatment indicated that TfR1-dependent iron uptake was selectively required for the maintenance of intestinal Treg cells in vivo.

More importantly, TfR1-dependent Treg cells were critical for the maintenance of immune tolerance in the intestine. As early as three weeks after the initial tamoxifen treatment, *Tfrc* iKO mice showed signs of colitis, including inflammatory cell infiltration (Supplementary Fig. S8g), slower body weight gains (Fig. 4h), and shortening of the colon (Fig. 4 i, j), while no obvious inflammation was observed in other organs (e.g., lung, liver, and kidney, Supplementary Fig. S8g). These results together highlight the importance of TfR1-dependent Treg cells for the maintenance of immune tolerance in the intestine.

### Microbiota-produced pentanoate enhances iron uptake and promotes c-Maf⁺ Treg differentiation

The higher levels of labile iron and the selective dependence of TfR1-mediated iron uptake indicate that intestinal Treg cells may procure iron via unique sources. Consistent with previous findings on intestinal epithelial cells in germ-free mice[50] and rats[51], we found that the microbiota in the intestine promoted iron uptake by Treg cells. Treg cells from either the colons or spleens of germ-free mice contained less labile iron than their counterparts in specific pathogen-free (SPF) mice (Fig. 5a). Expression of ferritin, a surrogate marker of intracellular iron levels[52], was also decreased in colonic Treg cells from germ-free mice (Fig. S9a). More importantly, although the Treg cell numbers in the colons of germ-free mice were decreased compared to those in the colons of SPF mice, as reported[53], treatment with iron dextran increased the levels of cellular iron and subsequently restored the colonic Treg cell ratios in the germ-free mice (Fig. 5b, c).

The importance of microbiota-facilitated iron uptake for intestinal Tregs was further validated in microbiota-depleted SPF mice. Pan antibiotic (ABX) treatment depleted the intestinal microbiota and reduced the levels of labile iron in colonic Treg cells (Supplementary Fig. S9b). As reported, antibiotic treatment diminished the Treg cells in the colon[53–56], but iron dextran treatment enhanced the cellular iron levels and subsequently restored the frequencies of colonic Treg cells in these mice (Supplementary Fig. S9 c,d). These data thus indicate that the commensal microbiota enhanced the iron absorption of Treg cells, which subsequently promoted the expansion of colonic Treg cells.

Metabolites produced by the commensal microbiota can shape the mucosal immune system[57,58]. To identify unique metabolites produced by the intestinal microbiota that could enhance iron uptake in Treg cells, we extracted intraluminal fecal contents from the cecum using the methanol-chloroform-water extraction method and compared the compositions of the inorganic phase between SPF mice and pan antibiotic-treated mice using gas chromatography coupled with mass spectrometry (GC–MS)[59]. The levels of two SCFAs, pentanoate acid and butyrate acid, were reduced most extensively in the antibiotic-treated mice (Fig. 5d). Similar to butyrate acid[60], pentanoate acid was absent in the colon of germ-free mice, further confirming their microbial origins (Fig. S9e).

Further analysis revealed that pentanoate promoted iron procurement and subsequently induced HIF-2α and c-Maf expression in Treg cells. Although both butyrate[60] and pentanoate enhanced iTreg differentiation in vitro (Supplementary Fig. S9f), only pentanoate increased the levels of intracellular iron (Supplementary Fig. S9g), promoted HIF-2α 5′-UTR activity (Supplementary Fig. S9h) and enhanced the expression of HIF-2α (Supplementary Fig. S9g) and c-Maf (Supplementary Fig. S9i, j) in iTreg cells. Such activities of pentanoate were blocked by the iron chelator DFO, confirming that pentanoate functions by enhancing iron uptake (Supplementary Fig. S9 i, j. Similar to these in vitro findings, pentanoate boosted iron uptake, increased the levels of intracellular iron (Fig. 5e, f left panels), and increased the expression of c-Maf (Fig. 5e, f middle panels) and HIF-2α (Fig. 5e, f, right panels) in colonic Treg cells of microbiota-depleted mice. Consequently, the ratios of colonic Treg cells in the antibiotic-treated mice were partially restored following pentanoate treatment (Fig. 5g, h).

Taken together, our data indicate that microbiota-produced pentanoate promoted the homeostasis of intestinal Treg cells by boosting iron uptake and subsequently enhancing the expression of c-Maf and HIF-2α in Treg cells.

### Pentanoate alleviates both iron deficiency and intestinal inflammation in DSS-induced colitis

Iron deficiency and the resulting anemia are currently poorly managed in IBD patients[61]. The ability of pentanoate to promote both iron uptake and colonic Treg homeostasis may provide strategies to alleviate intestinal inflammation as well as iron-deficient anemia in subjects with colitis. In a murine model of acute colitis induced by dextran sodium sulfate (DSS) (Fig. 6a), the red blood cell, hemoglobin, and hematocrit values were severely diminished as early as 7 days after DSS treatment (Fig. 6b). In contrast, when pentanoate was administered to the mice prior to DSS treatment, the iron deficiencies and resulting anemia were substantially mitigated, as these mice exhibited increased serum iron levels (Fig. 6c), increased red blood cell counts, and enhanced hemoglobin and hematocrit levels compared to those of mock-treated animals (Fig. 6b). Notably, many signs of colitis, including shortened colon (Fig. 6d, e) and intestinal inflammation (Fig. 6f, g), were significantly alleviated following pentanoate treatment.

More importantly, pentanoate also alleviated intestinal inflammation and iron deficiencies in mice with established active colitis. A murine model of chronic colitis was induced by repeated cycles of dextran sodium sulfate (DSS) (Fig. 7a). In such mice with ongoing colitis, administration of pentanoate greatly mitigated the iron deficiencies and resulting anemia, including increased red blood cell counts, and enhanced hemoglobin and hematocrit levels compared to those of mock-treated animals (Fig. 7b). Shortened colon (Fig. 7c, d) and intestinal inflammation (Fig. 7e, f) also significantly improved following the pentanoate treatment. These data thus suggest that pentanoate could be an ideal agent that mitigates both iron deficiency and colonic inflammation in subjects with colitis.

Further supporting the importance of pentanoate in iron homeostasis and intestinal inflammation, levels of pentanoate in stool samples were reduced in patients with either active Crohn's disease (CD) or ulcerative colitis (UC) in both HPM2 and PRISM cohorts compared to control individuals[62,63] (Fig. 7g, h, Supplementary Data 1, 2). Expression of cellular Ferritin heavy chain (FTH1), a surrogate marker for the intracellular iron level[52], was diminished in patients with CD or UC (Suppl. Fig.S9i), and more importantly, was positively correlated with levels of pentanoate in HPM2 cohorts (Fig. 7i).

### Discussion

In this study, we demonstrate that the differentiation of intestinal Treg cells stringently relied on their higher levels of intracellular iron, which promoted HIF-2α and c-Maf expression and was critical for the

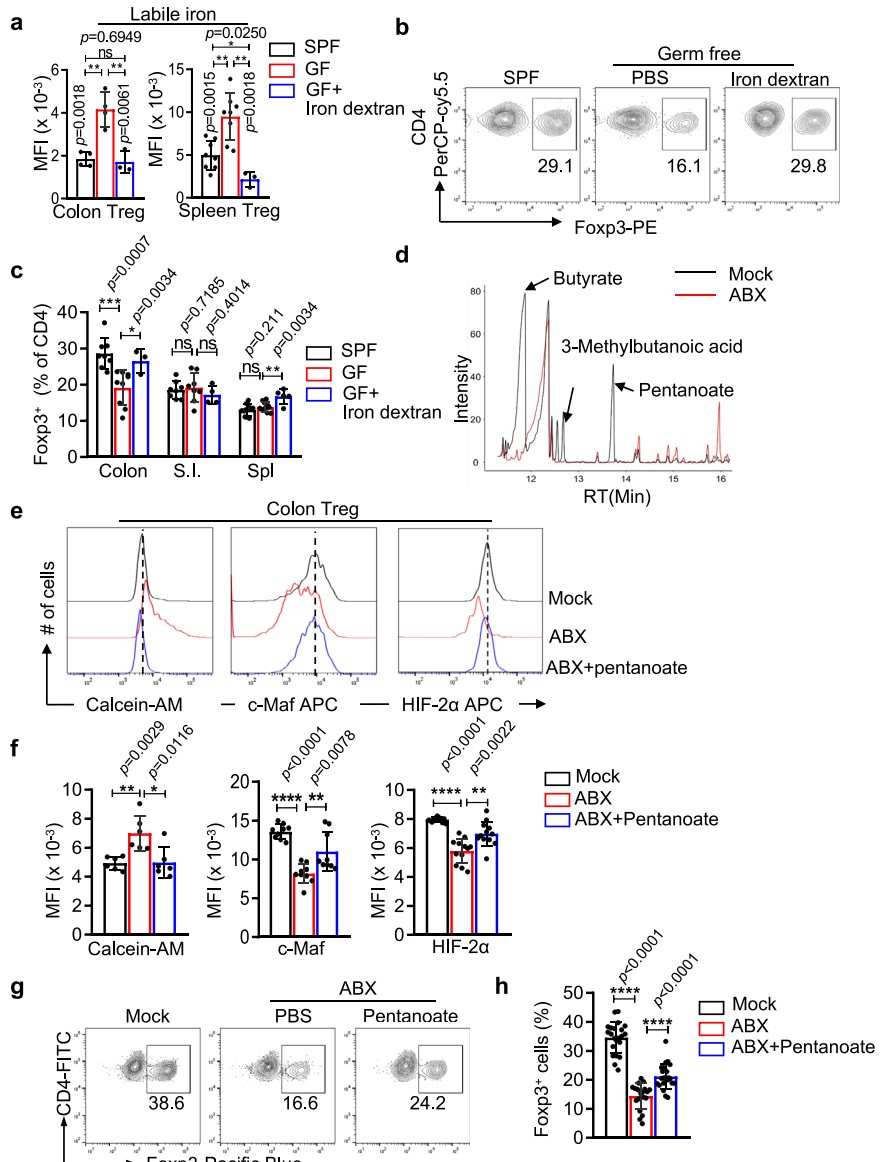

**Fig. 5 | Pentanoate produced by microbiota promotes iron uptake and intestinal Treg differentiation. a** Treg cells in the colon (*left*) and spleen (*right*) were isolated from specific pathogen-free mice (SPF), germ-free (GF) and iron-dextran-treated germ-free mice. Labile iron levels were determined by staining with calcein-AM, the fluorescence of which is reversely correlated with levels of intracellular labile iron. Colon, $n = 4,4$, 3 mice for SPF, GF, GF+iron dextran group, respectively, Spleen, $n = 8$, 83 mice for SPF, GF, GF+iron dextran, respectively. **b, c** Germ-free mice were treated with iron dextran and Tregs in the colon were assessed by Foxp3 intracellular staining. PBS-treated germ-free mice and SPF mice were included as controls. **b** Representative flow cytometry profiles of pregated CD4$^+$CD45$^+$ cells from the colon; (**c**) Summary of the percentages of Foxp3$^+$ cells in the indicated organs (colon, $n = 8,9$, 3 mice for SPF, GF, GF+iron dextran group, respectively; S.I., $n = 8,8$, 4 for SPF, GF, GF+iron dextran group, respectively; spleen, $n = 11,11,5$ for SPF, GF, GF+iron dextran group, respectively). **d** Metabolites from the caeca of SPF (mock) and antibiotic-treated SPF (ABX) mice were analyzed by gas chromatography–mass spectrometry (GC–MS). The data are representative of two independent experiments with a total of four mice in each group. **e, f** SPF mice were treated with pan antibiotics (ABX) and administered with sodium pentanoate via their drinking water for 21 days (ABX+pentanoate). The levels of labile iron, c-Maf and HIF-2α in colonic Tregs were determined by FACS. Untreated SPF mice (mock) and antibiotic-treated SPF mice (ABX) were included as controls. Representative (**e**) or summarized (**f**) levels of labile iron (Calcein-AM staining), c-Maf and HIF-2α from three independent experiments are shown (Calcein-AM staining, $n = 6$ mice; c-Maf staining, $n = 9$ mice; HIF-2α staining, $n = 12$ mice). **g, h** The ratios of colonic Treg cells were determined by intracellular staining. **g** Representative flow cytometric analysis; (**h**). Summary of four independent experiments ($n = 22, 20, 22$ micefor SPF, ABX, ABX+pentanoate, respectively). **a, c, f, h** Statistical significance was determined by a two-tailed *t* test. *$p < 0.05$, **$p < 0.01$, ***$p < 0.001$, ****$p < 0.0001$. Data shown are mean ± SD.

maintenance of immune tolerance in the intestine. Importantly, increased intracellular iron levels in intestinal Treg cells were dependent on microbiota-produced pentanoate, and either iron or pentanoate rescued defects in intestinal Treg cells in microbiota-depleted mice. Our study thus revealed a mechanism by which the microbiota promotes intestinal Treg cell homeostasis by modulating nutrient procurement.

Our results indicate that pentanoate could serve as a therapeutic to simultaneously restore iron deficiencies and immune tolerance in patients with IBD. Although direct iron supplementation is currently used to alleviate anemia associated with IBD, global iron excess often leads to iron-mediated oxidative stress and overgrowth of pathogenic bacteria in the intestine[64]. As demonstrated in the DSS-induced colitis model, the unique abilities of pentanoate to promote iron

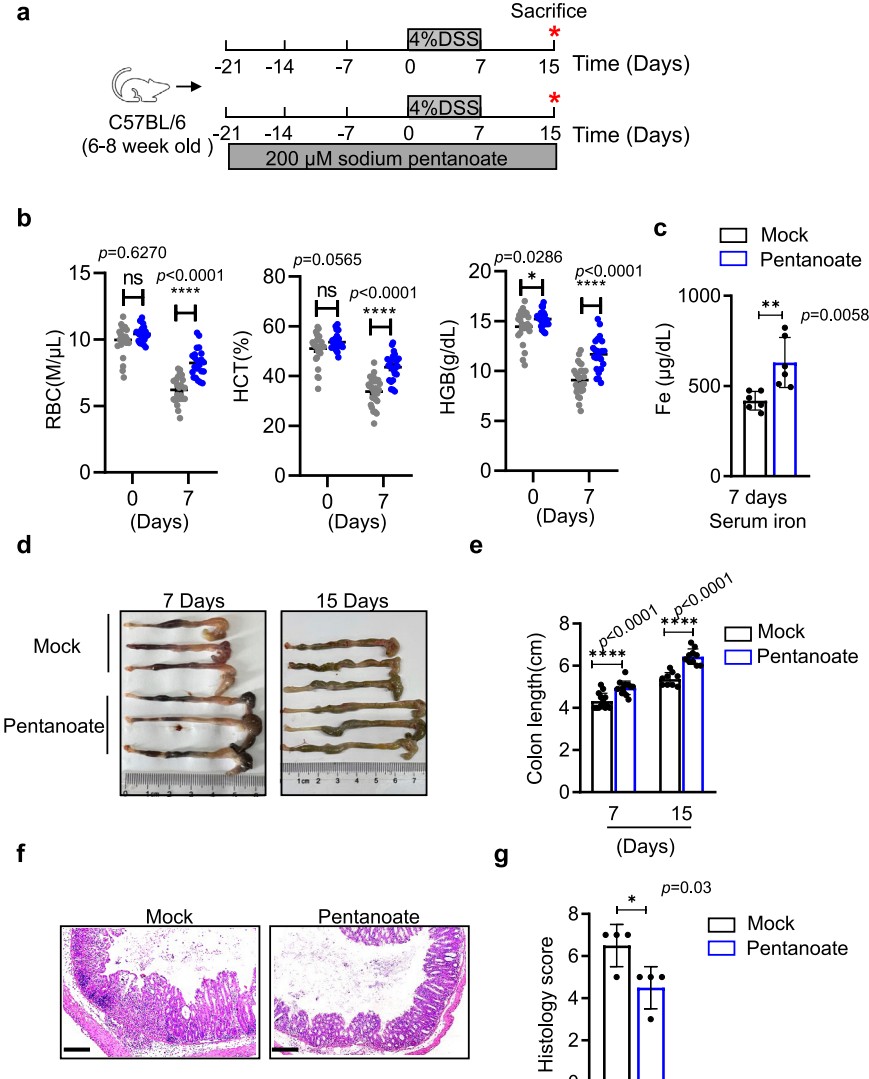

**Fig. 6 | Pentanoate prevents colitis-associated anemia and intestinal inflammation. a** Schematic of the experiments used to evaluate sodium pentanoate in DSS-induced colitis. Briefly, six-week-old C57BL/6 mice were treated with sodium pentanoate (200 mM) via their drinking water for 3 weeks and then further administered DSS (4%) for seven days. **b** Seven days after DSS treatment, erythroid parameters, including the red blood cell counts and hematocrit and hemoglobin levels were determined with a hematology analyzer (ProCyteDx). Statistical significance was determined by one-way ANOVA ($^*p < 0.05$, $^{****}p < 0.0001$, $n = 26$ mice examined over three independent experiments). Data are presented as the mean ± SEM. **c** as in (**b**), serum iron levels were determined by ICP-MS (iCAP RQ, Thermo Fisher). For ICP-MS detection, $n = 6$ mice examined over two independent experiments. **d**–**g** The lengths **d**, **e** and hematoxylin and eosin staining **f**, **g** of colons on days 7 and 15 after DSS challenge were shown. Scale bars represent 200 μm. Data are from three independent experiments (**e**, $n = 14$ and 13 mice for 7 days after mock and pentanoate treatment respectively; $n = 9$ and 10 mice for 15 days after mock and pentanoate treatment respectively; **g**, $n = 4$ mice per group). **c**, **e**, **g** Statistical significance was determined by two-tailed Student's $t$ test. $^*p < 0.05$, $^{**}p < 0.01$, $^{***}p < 0.001$, $^{****}p < 0.0001$. Data shown are mean ± SD.

procurement and Treg homeostasis alleviated both iron deficiencies and intestinal inflammation. Pentanoate is produced by intestinal commensal bacteria such as *Megasphaera massiliensis*[65]; thus, either direct pentanoate supplementation or modulation of *Megasphaera messiliensis* could be helpful for improving both the enteric conditions and nonenteric conditions of patients with IBD. It's worth noting that IBD is a heterogenous disease and DSS-induced colitis in mice does not recapitulate all the features of IBD in humans.

Several mechanisms may contribute to the unique capacity of pentanoate to improve iron absorption. As an SCFA, pentanoate may increase the solubility and availability of iron in the colon[66,67]. Alternatively, pentanoate may function via the SCFA receptor FFAR2 (GPR43) to modulate the expression of genes related to iron homeostasis (e.g., TfR1). Last, as a potent HDAC inhibitor[65], pentanoate may also stimulate the expression of

genes involved in iron absorption, further supporting its role in iron homeostasis. In addition to its activities in promoting iron procurement, pentanoate promoted IL-10 expression in T cells and B cells in vitro[60]. Given that IL-10 is a well-established target of c-Maf in various immune cells[41,60,68–70], further studies are needed to determine whether pentanoate promotes IL-10 via the HIF-2α-c-Maf axis in lymphocytes other than Treg cells.

The intestinal microbiota is critical for nutrient procurement as well as for the maintenance of immune tolerance. In addition to the initially proposed "iron tug war" between the microbiota and the host in competing for iron by secreting siderophores[71], the gut microbiota also modulates host iron uptake in the intestine[28] by increasing iron bioavailability, modulating the expression of genes involved in iron uptake[7], and secreting cofactors to facilitate iron transport[72]. Notably, in contrast to iron-replete conditions, the microbiome may inhibit iron

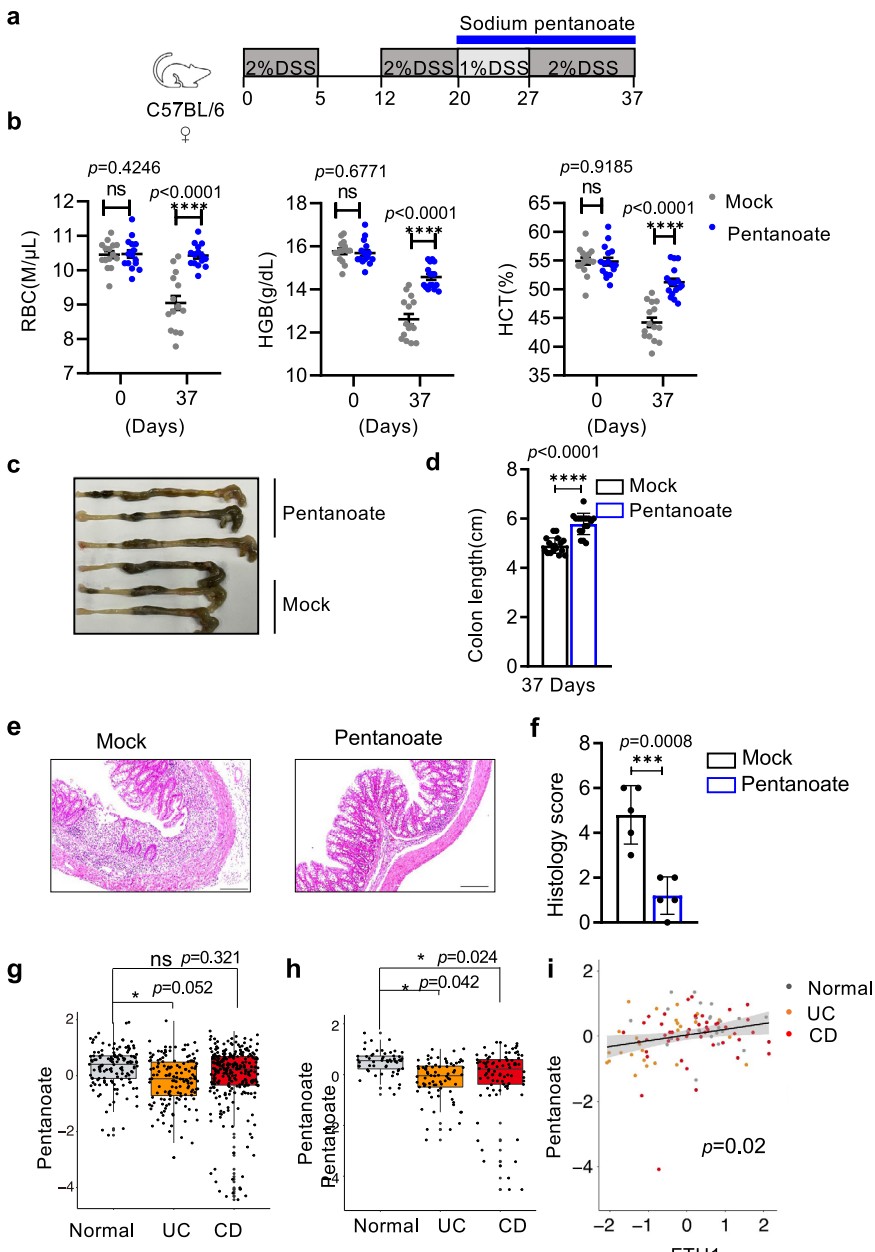

**Fig. 7 | Pentanoate improves anemia and intestinal inflammation associated with established colitis. a** Schematic of the experiments to evaluate pentanoate in chronic colitis after the disease onset. Briefly, WT mice were administrated 2% DSS for five days, followed by seven days of water or 1% DSS for 3 cycles. Twenty days later, 200 mM sodium pentanoate or vehicle was added to the drinking water, and the mice were analyzed 17 days after the pentanoate treatment. **b** As in (**a**), blood were collected and erythroid parameters, including the red blood cell (day 0, $n = 15$ and 16 mice for mock and pentanoate treatment group; day 37, $n = 15$ mice), HGB and hematocrit ($n = 15$ and 16 for mock and pentanoate treatment group respectively) were measured over three independent experiments. Error bars represent standard errors of the mean, ****$p < 0.0001$ in one-way ANOVA Test. **c–f** As in (**a**), the lengths and H&E staining of colons were determined. **e** Scale bars represent 200 μm. The data are representative (**c, e**) or summary (**d, f**) of two independent experiments. **d** $n = 17$ mice in each condition, **f** $n = 5$ mice. Statistical significance

was determined by two-tailed Student's $t$ test. ***$p < 0.001$. Data shown are mean ± SD. **g, h** Pentanoate levels in stool samples from patients with Crohn's disease (CD), Ulcerative colitis (UC), and control individuals (normal) were determined with LC-MS. Data are Z-score transformed based on previously published HPM2 (**g** $n = 146$, $n = 265$, $n = 135$ for individuals with UC, CD or without IBD) or PRISM (**h**, $n = 88$, $n = 76$, $n = 56$ for individuals with CD, UC, or without IBD) studies[62,63]. Box plots indicate median and interquartile range (IQR). The upper and lower whiskers indicate 1.5 times the IQR from above the upper quartile and below the lower quartile.*$p < 0.05$; *ns*, not statistically significant based on linear regression analysis. **i** As in **g**, pentanoate levels in the stool samples were positively correlated with FTH1 expression (FPKM), a surrogate for intracellular iron levels, in the intestine determined with RNA-seq. Data are Z-score transformed based on HPM2 studie[63]. $p = 0.02$ based on Pearson coefficient analysis (two-sided). The error band represents the linear regression line with a 95% confidence interval.

absorption under iron-deficient conditions[7] via lactobacillus-produced 2,3-diaminopropionic acid (DAP) and reuterin to inhibit iron uptake[73]. Thus, microbial-dependent iron homeostasis may play a pivotal role in the interplay between nutrient procurement and immune tolerance under different physiological settings.

In addition to the differentiation of intestinal Treg cells, intracellular iron is indispensable for the effector function of Treg cells. Notably, both HIF-2α and c-Maf were recently identified to be essential for the effector program and suppressive functions of Treg cells, although the severity of autoimmune disease in TfR1 cKO mice was

much greater than that in mice with either c-Maf-deficient[29,30] or HIF-2α-deficient Tregs[74]. The functions of iron beyond promoting HIF-2α-c-Maf expression may also contribute to the defective effector activities of Tregs and lethal autoimmunity in *Tfrc* cKO mice. Similar to the importance of iron for conventional T-cell proliferation[75], the proliferation and activation of Treg cells were impaired due to partial iron deficiency caused by TfR1 deficiency. Furthermore, given the critical role of iron in mitochondrial activities[75] and the roles of mitochondria in Treg effector functions[76,77], defective mitochondrial functions in TfR1-deficient Treg cells may contribute to the loss of Treg suppressive activity and lethal autoimmunity observed in TfR1 cKO mice. Further studies are warranted to elucidate the iron-dependent effector functions of Tregs.

It's worth noting that iron-mediated signaling plays a complex and multifaceted role in the regulation of many biological processes. In addition to its roles in post-transcriptional regulation via IREs, iron can bind to enzymes (e.g. *JmjC* demethylases[15], prolyl hydroxylase domain enzymes[78]) involved in gene expression and directly to certain transcription factors[79] to modulate their activities. Future research on roles of iron-mediated signaling in the immune system may provide further opportunities for the treatment of diseases associated with immune dysfunctions.

## Methods

### Mice

*Tfrc*[fl/fl] mice were obtained from National Resource Center of Model Mice (Nanjing, China) and backcrossed to C57BL/6 background for over ten generations. *Foxp3*[YFP-IRES-Cre], *Foxp3*[eGFP-creERT2], *Rag1*[-/-], and C57BL/6 SJL (CD45.1) mice were obtained from Jackson laboratory (Bar Harbor, ME). All mice were maintained under specific pathogen-free conditions with free access to a standard mouse chow (JXT, Jiangsu Xietong) and sterile water unless otherwise specified. The mice were housed in a controlled environment with a 12-h light/dark circle and temperatures ranging from 21.1–22.2 °C. The humidity level was maintained at 40%-60%.

Germ-free mice were obtained and housed in germ-free isolators from National Resource Center of Model Mice (Nanjing, China). To acutely deplete TfR1 in Treg cells, *Foxp3*[eGFP-creERT2] mice were treated with 320 mg tamoxifen /kg (body weight) by oral gavage every four days for total four times. All animal studies were performed in compliance with the guide for the care and use of laboratory animals and were approved by the institutional biomedical research ethics committee of the Westlake University. Sex was not considered in the study design.

### Antibiotics treatment

6-week-old C57BL/6 mice were treated initially with a mixture of antibiotics (5 mg metronidazole, 2.5 mg vancomycin, 5 mg ampicillin, and 5 mg neomycin) by oral gavage every day for 5 days. An antibiotic cocktail (1 mg/mL metronidazole, 0.5 mg/mL vancomycin, 1 mg/mL ampicillin, and 1 mg/mL neomycin) was also supplemented in the drinking water for an additional three weeks.

### Iron dextran treatment

For neonatal mice, iron dextran (Sigma) was first injected intraperitoneally at postnatal day 3 (5 mg/e.a.) and day 9 (7.5 mg/e.a). The mice were then continuously treated with iron dextran (12.5 mg /e.a.) once a week until postnatal day 44. For germ-free mice, iron dextran was intraperitoneally injected (37.5 mg/e.a.) every 5 days for 3 weeks.

### Analyzing the activity of Pentanoate in a DSS-induced colitis model

Six-week-old C57BL/6 mice were supplemented with 200 mM sodium pentanoate in their drinking water three weeks prior to colitis induction. Colitis was induced by adding 4% (w/v) DSS (molecular weight

36–50 kDa; MP Biomedicals) to the drinking water for seven days. Blood was collected on day 0 and day 7 following DSS treatment using BD Microtainer tubes containing EDTA anticoagulant, and erythroid parameters were measured with a Hematology Analyzer (ProCyteDx). To determine the severity of colitis, colon samples were fixed overnight in 10% neutral-buffered formalin and stained with hematoxylin and eosin. Histological evaluations were assessed in a double-blinded fashion based on three parameters: damage to crypt architecture, edema in the sub-mucosa, and infiltration of inflammatory cells.

### Analyzing the activity of pentanoate in an established colitis model

Chronic colitis was induced by administering 2% (w/v) DSS (molecular weight 36 KD-50 KD; MP Biomedicals) in drinking water for five days, followed by seven days of water or 1% DSS for three cycles. After a 20-day period, 200 mM sodium pentanoate was administered, and erythroid parameters were measured using a Hematology Analyzer (ProCyteDx) on the 17th day following the administration of sodium pentanoate or vehicle.

### Gas Chromatography-Mass Spectrometry analysis of metabolites

Metabolites from mouse cecum contents were extracted in both organic and aqueous phases using the methanol-chloroform-water extraction method. Briefly, 3 mL of pre-cooled methanol/chloroform (2:1, v/v) (Sigma) was added to a tube containing freshly collected samples, which were pre-weighed. After homogenization with an Ultrasonic Processor, the samples were centrifuged at 8000 $g$ for 10 min at 4 °C, and the supernatant was collected into tubes containing 2 mL ice-cold water. After vigorous vortexing, the sample tubes were centrifuged at 18,000 $g$ for 10 min at 4 °C to obtain phase separation. The upper phases were collected without disturbing the interface and were dried using a Speedvac (Eppendorf Vacufuge) to obtain a metabolite pellet.

Metabolites extracted from cecal contents were dissolved in deionized water and homogenized with an extraction solution containing 50 μL HCl and 200 μL ether. After vigorous shaking, samples were centrifuged at 1,000 $g$ for 10 min, and the top ether layer was collected. 80 μL of the ether extracts were mixed with 16 μL N-tert-butyldimethylsilyl-N-methyltrifluoroacetamide (MTBSTFA), heated at 80 °C for 20 min, and left at room temperature overnight for derivatization. The derivatized samples were run through a Gas Chromatography-Mass Spectrometry (GC-MS) (5977B GC/MSD, Agilent Technologies) equipped with an HP-5MS column (0.25 mm × 30 m X 0.25 μm). Pure helium (99.9999%) was used as the carrier gas at a flow rate of 1.2 mL/min. The head pressure was set at 5 psi with a split ratio of 100:1. The inlet and transfer line temperatures were 250 and 300 °C, respectively. The following temperature program was used: 60 °C (3 min), 60–120 °C (5 °C/min), 120–300 °C (20 °C/min). Organic acids were identified using NIST MS Search 2.3.

### Lymphocyte staining and flow cytometry

For cell surface staining, cells were washed with FACS buffer (2% FBS in PBS with 1 mM EDTA), blocked with an antibody against CD16/32 (2.4G2, BD Pharmingen), and incubated with indicated surface antibodies on ice for 30 min. Cells were then washed two more times with FACS buffer and fixed in 1% paraformaldehyde in PBS before being analyzed with a CytoFLEX machine (Beckman). For intracellular cytokine staining, cells were stimulated with phorbol 12-myristate 13-acetate (PMA, Sigma) (50 ng/ml) +ionomycin (Sigma) (500 ng/ml) for 3 h in the presence of Golgi-stop (BD Bioscience). Cells were then surface stained, fixed/permeabilized with a Cytofix/Cytoperm kit (BD Bioscience), and stained with antibodies against indicated cytokines. Intracellular Foxp3 and c-Maf staining were carried out according to manufacturer's instruction (Ebioscience). For co-staining Foxp3 with

YFP, cells were fixed by 3.7% formaldehyde (Sigma), permeabilized by 0.2% Triton X-100 and stained with indicated antibodies in the FACS buffer for 90 mins.

The following FACS antibodies were used in the study: TfR1-APC (1:200, R17217, Invitrogen, 17071182), CD44-APC (1:200, IM7, Biolegend, 103012), CD44-FITC (1:200, IM7, Biolegend,103005), CD62L-PE/Cy7 (1:200, MEL-14, Invitrogen, 25062182), Foxp3-PE (1:100, FJK-16s, Invitrogen, 12577382), CD4-PerCP/Cy5.5 (1:200, GK1.5, Biolegend, 100434), Foxp3-Pacific Blue (1:100, FJK-16s, Invitrogen, 48577382), CD25-APC (1:200, PC61, Biolegend, 102012), c-Maf-APC (1:200, symOF1, Invitrogen, 50985582), HIF-2α-APC (1:100, ep190b, Invitrogen, MA5-16021), HIF-1α-APC (1:100, Mgc3, Invitrogen, 17-7528-82), anti-CD16/32 (1:400, 2.4G2, BD Pharmingen™, 553142), IL4-PE/Cy7 (1:200, 11B11, Biolegend, 504117), IL-17A-APC (1:200, TC11-18H10.1, Biolegend, 506911), INF-γ-APC (1:200, XMG1.2, Biolegend, 505810), CD4-FITC(1:200, GK1.5, BD Pharmingen™, 557307), CD11b-PerCP/Cy5.5 (1:200, M1/70, Biolegend,101228), CD19-PE/Cy7 (1:200, 6D5, Biolegend,115520), CD8-APC (1:200, 53-6.7, Biolegend, 100724), CD4-Pacific Blue (1:200, GK1.5, Biolegend,100428), CD44-PE (1:200, IM7, Biolegend, 103024), OX-40-PE/Cy7 (1:200, OX-86, Biolegend, 119415), GITR-APC (1:200, DTA-1, Biolegend, 126312), PD-1-APC (1:200, J43, eBioscience™, 17998582), Ki-67-APC (1:200, 16A8, Biolegend, 652406), CD152-PE/Cy7 (1:200, UC10-4B9, Biolegend, 106314), Helios-PE/Cy7 (1:200, 22F6, Biolegend, 137236), ICOS-APC (1:200, 7E.17G9, Invitrogen, 17994282), CD103- PE/Cy7 (1:200, 2E7, Biolegend, 121425), Ahr-PE (1:100, 4MEJJ, Invitrogen, 12-5925-82).

To detect liable iron, the mononuclear cells were washed with PBS twice and subsequently treated with 1 mM calcein-AM (Invitrogen) for 10 min at 37 °C. Following this, the cells were labeled with flow cytometric antibodies as described previously.

### Isolation of intestinal lamina propria cells
Intestines were collected and incubated in pre-digestion buffer (1 mM DTT, 10 mM EDTA and 10 mM HEPES in PBS) at 37 °C for 30 min to remove epithelial cells. Remaining tissues were dissociated in digestion buffer containing 1 mg/mL collagenase D (Worthington), 20 μg/mL DNase I (Roche), and 10% FBS (Hyclone) with constant stirring at 37 °C for 30 min. Mononuclear cells were then collected at the interface of a 40–70% Percoll gradient (GE Healthcare).

### Cell purification and culture
To purify naïve CD4 T cells, splenocytes and lymph node cells were purified using mouse naïve CD4⁺T cell isolation kit (Stem cell) according to manufacture instructions. Purity of cells was routinely above 95%. Purified naïve CD4 T cells were stimulated with anti-CD3 (3 μg/mL; 2C11) and anti-CD28 (2 μg/mL; 37 N) in T cell medium (IMDM, 10% FBS, 1x antibiotics, 2 mM L-Glutamine and 50 μM β-mercaptoethanol). To induce iTreg differentiation, following blocking antibodies and recombinant cytokines were added into the culture: anti-IL4 (5 μg/mL), anti-IFNγ (5 μg/mL; R46A2), rhIL2 (100 U/mL, Peprotech) and 5 ng/mL rhTGF-β (RD systems). When indicated, iTreg cells were stimulated with 50 μM FeSO4(Sigma), 0.1 mM sodium butyrate (Sigma), 1 mM sodium pentanoate (Macklin), or 50 μM PT2385 (Selleck).

To analyze cell proliferation, CD4 T cells were purified and labeled with CellTrace Violet reagent (Life Tech) and stimulated under iTreg condition.

### ICP-MS analysis
To measure iron in Treg cells of TfR1 cKO and WT mice, CD25⁺YFP⁺ Treg cells from *Tfrc^{fl/fl} Foxp3 ^{YFP-Cre/+}* or *Tfrc^{+/+} Foxp3 ^{YFP-Cre/+}* female mice were sorted in iron-free tubes and lysed in ultra-pure HNO3 (Thermo) for 1 h at 95 °C. The lysate was further diluted with DI water and subjected to metal analysis with ICP-MS (iCAP RQ, Thermo fisher).

For iron analysis in PBMC, serums were diluted 20-fold with DI water including 0.1% Triton X 100 and 0.1% HNO3 and analyzed by ICP-MS.

### Luciferase assay in Treg
The luciferase constructs for HIF-2α 5′-UTR IRE were previously reported (Mayka Sanchez et al., 2007). Naïve CD4 T cells were purified and cultured in the iTreg differentiation condition. 72 h after the differentiation, 2.5 million cells were transiently transfected with 9 μg pGL3 (HIF-2α-5′UTR)-Luc and 0.5 μg renilla luciferase reporter using a mouse primary T cell nucleofection kit (Lonza AG). After resting for 12 h, the cells were cultured in 750 μL Treg differentiation medium with either 50 μM FeSO4, 1 mM sodium pentanoate or PBS for additional 12 h. Cell lysate was prepared, and activity of firefly luciferase (FL) and Renilla luciferase (RL) was determined with a dual-luciferase reporter assay (Promega), and FL activity was normalized to RL activity.

### RNA extraction, cDNA synthesis and quantitative Real-Time PCR
Total RNA was isolated with RNeasy plus kit (Qiagen, CA) and cDNA was synthesized using PrimeScript™ RT reagent Kit with gDNA Eraser (TaKara, Japan). Quantitative PCR was performed using SYBR Premix Ex Taq (TaKara, Japan) on a StepOnePlus real-time PCR system (Life Tech, USA). Gene expression was normalized to 18 S rRNA or β-actin using the ΔΔCt method. Primer sequences were listed in Supplementary data 3.

### RNA seq analysis
CD4 T cells from *Tfrc^{fl/fl} Foxp3 ^{YFP-Cre/+} or Tfrc^{+/+} Foxp3 ^{YFP-Cre/+}* female mice were first isolated and stained with antibodies against CD4 (BioLegend, GK1.5), CD25 (BioLegend, PC61), and CD25⁺YFP⁺ Treg cells were sorted by flow cytometry (BC MoFlo Astrios, Beckman), then cells were lysed to extract RNA. RNA quality and quantity were measured using Agilent Bioanalyzer. mRNA libraries were prepared using NEBNext® Single Cell/Low Input RNA Library Prep Kit (E6420, NEB). DNA libraries were sequenced on a Hiseq 2000 platform or Hiseq X-10 platform. To analyze the transcriptome change caused by iron supplement, WT Treg cells were treated with FeSO4 (50 μM) or PBS for 4 days. Two pairs of samples from two independent experiments were included for analysis. RNA seq reads were first mapped to mm9 with STAR[80], and differential gene expression (RPKM) analysis was conducted by Deseq. Heat map was generated with Cluster3 and Java Treeview[81], and genes and samples were clustered with hierarchical clustering. Gene ontology analysis was done using Metascape. GSEA analysis used samples expression dataset (FPKM matrix) and all gene sets (v7.5) as input. Gene sets at an FDR less than 10% were chosen for display and the normalized enrichment score (NES) was reported.

### Single cell RNA seq
Droplet-based scRNA-seq datasets were produced using a Chromium system (10x Genomics), referred to as 10x. Treg cells (CD4⁺CD25⁺) from 6-week-old *Tfrc^{fl/fl} Foxp3 ^{YFP-Cre/+}* female mice were sorted as above, manually counted, and their concentrations adjusted to enable the capture of 10000 cells, the standard protocol for 10x Single Cell 3′ Reagent Kit (V2 chemistry) was followed. Libraries were sequenced on the NovaSeq 6000 platform.

### Bone marrow transplantation
Bone marrow cells were purified from 6-day-old *Tfrc ^{flox/flox} Foxp3 ^{Cre-IRES-YFP}* or *Tfrc ^{+/+} Foxp3 ^{Cre-IRES-YFP}* mice, and infiltrating T cells were depleted using anti-CD4 and anti-CD8 antibodies and magnetic beads (Stem Cell). Bone marrow cells were then mixed with congenic WT bone marrow cells (CD45.1⁺), and five million cells were injected intravenously into *Rag1^{-/-}* mice which received sub-lethal irradiation (400 rad) one day before the

transplantation. Chimeric mice were analyzed eight weeks after transplantation.

### Adoptive transfer

CD45.1$^+$CD4$^+$CD45RB$^{high}$CD25$^-$ (Teff) were sorted with flow cytometry from wild-type C57BL/6 mice. CD45.2$^+$CD4$^+$YFP$^+$ (Treg) cells were sorted from either 6-day-old *Tfrc* $^{flox/flox}$ *Foxp3* $^{Cre-IRES-YFP}$ or *Tfrc* $^{+/+}$ *Foxp3* $^{Cre-IRES-YFP}$ mice. 400,000 Teff cells together with 100,000 Treg cells mice were mixed and injected into Rag-1-deficient mice. Recipient mice were weighed at the time injection and 3 times a week for 12 weeks or until the mice lost 20% of their initial body weight.

### Retroviral infection

Retroviruses were produced by transfecting Plat-E cells with the indicated plasmids. Fresh virus supernatant was collected 2 days after transfection and used to infect Treg cells 18 and 40 h after activation in the presence of polybrene (8 µg/mL). After spin infection for 1.5 h at 900 g, Treg cells were cultured at 37 °C for an additional 2 h before being resuspended in the indicated Treg differentiation medium.

### Statistics & reproducibility

Statistical tests were performed using GraphPad Prism version 9.0, and all statistical parameters are reported in the figure legends. Differences between two or more groups were analyzed by either Student's *t*-test or ANOVA followed by Tukey's post-hoc test, as appropriate. Sample size was not predetermined by statistical methods, and no data were excluded from the analyses.

To evaluate the severity of colitis, H&E staining results were scored by a blinded pathologist. The investigators were not blinded to group allocation during the other experiments and outcome assessment. In experiments with established colitis, the mice were randomly assigned to treatment and control groups. The other experiments were not randomized.

### Re-Analysis of HPM2 and PRISM data

Two public IBD datasets (PRISM and HMP2) were analyzed to determine pentanoate abundance in stool samples of patients with Crohn's disease (CD), ulcerative colitis (UC), and non-IBD control individuals. For this study, we selected 220 patients diagnosed with CD ($n = 88$), UC ($n = 76$), and non-IBD ($n = 56$) conditions from the PRISM dataset. The HMP2 dataset contained 546 individuals with CD ($n = 265$), UC ($n = 146$) and control individuals without IBD ($n = 135$). Stool samples were analyzed with metabolomic profiling using a combination of four LC−MS methods. Participants classified as having CD or UC were compared with control (non-IBD) individuals. Specifically, in HMP2 cohorts, a linear mixed-effect model was applied to determine pentanoate levels, with age as a continuous covariate, and four medications (antibiotics, immunosuppressants, chemotherapy, and probiotic) as binary covariates. Pentanoate levels in the PRISM cohort were evaluated following a linear regression model, and age at study consent was included as a continuous covariate, and medications (antibiotics, mesalamine, immunosuppressants, and steroids) as a binary covariate. Additional host transcriptomes (HTX) from the HMP2 were further used to associate with pentanoate abundance, based on 254 HTX samples from 90 individuals (CD, $n = 43$; UC, $n = 25$; without IBD, $n = 22$). Initial normalization was conducted on raw sample-by-gene HTX count data using the voom method implemented in R's limma package. Normalized counts were used as a basis for linear mixed-effect modeling to detect differential gene expression, controlling for age, sex, and biopsy location. Spearman correlation was used to associate HTX expression with pentanoate status. All statistical tests were performed using GraphPad Prism version 9.0, and statistical parameters are reported in the figure legends.

### Reporting summary

Further information on research design is available in the Nature Portfolio Reporting Summary linked to this article.

## Data availability

The high-throughput sequencing data generated in this paper has been deposited in the GEO under accession number GSE195607 (https://www.ncbi.nlm.nih.gov/geo/query/acc.cgi?acc=GSE195604). Taxonomic and functional profiles for HMP2 metabolomes (MBX) and host transcriptomes (HTX) were downloaded from http://ibdmdb.org in July 2020. Mass spectra data in PRISM were downloaded from metabolomics workbench study ST000923. Source data are provided as a Source Data file with the paper. Source data are provided with this paper.

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

## Acknowledgements

We thank Zongyi Zhang, Chao Zhang, and Youhui Yang for the facility support, technical assistance and discussion. We thank Dr. Changchun Xiao, Dr. Yichun Xiao and Dr. Zhizhang Wang for sharing critical reagents and providing suggestions. We thank advanced biomedical technology core facility, supercomputing center, Instrumentation and Service Center for Molecular Sciences and laboratory animal resource center at Westlake University for the facility support and technical assistance. This work was supported by Key R&D Program of Zhejiang Province (2022SDXHDX0002) (XC); National Key R&D Program of China (2022YFA0807300,2018YFA0107500, 2018YFA0801400); the National Natural Science Foundation of China (32025016,31870927, 81671552 & 81830078). This project is supported in part by Westlake Education Foundation and Tencent Foundation.

## Author contributions

L.Z. performed experiments, analyzed data, and wrote the manuscript. G.L, Z.L., T. Q, K.D., J. Y., Y. P., J. Z. and Y.S. assisted with the preparation of reagents, data analysis, and manuscript preparation. X.C. conceptualized the project, designed and supervised the research, and wrote the manuscript.

## Competing interests

L.Z and X.C. are coinventors on a patent application that have been filed by Westlake University on methods and reagents that target TfR1 and iron metabolism to modulate Treg cell differentiation. The other authors declare no competing interests associated with the study.
