## [Peer Review File · Nature Communications]

Microbiota-assisted iron uptake promotes immune tolerance in the intestineREVIEWER COMMENTS

Reviewer #1 (Remarks to the Author):

The current manuscript entitled 'Microbiota-assisted iron uptake promotes immune tolerance in the intestine' by Dr. Zhu and colleagues investigates fundamental biological functions of TfR1 in Tregs using KO models. The study is very comprehensive and well-conducted, and the data are novel and well presented.

The findings are relevant to basic immunology, and the authors demonstrate possible translational aspects in IBD.

I have several major and minor points that I would like to ask the authors to address:

Major points:

Fig. 1a

The authors claim that TfR1 was specifically deleted in Tregs in the Tfrc cKO mice. Can the authors provide the isotype control in Fig. 1a? The authors should also demonstrate TfR1 surface levels by flow cytometry on other cell types such as CD4+ T-bet+, CD4+ GATA3+, CD4+ RORgt+ and CD8+ cells in a supplementary figure.

I realize that some aspects are touched on in FigS1 but I still would like the authors to better characterize the phenotype of these Tfrc cKO mice:

What is the reason for the accumulation of CD4+ T cells in tissues? Is it increased proliferation or reduced cell death?

To better describe lymphocytic infiltration of organs, the authors should report the absolute numbers of CD4+, CD8+, CD19+ and CD56+ cells in the organs shown by histopathology in Fig. 1d.

Please also include cell numbers and cytology or histopathology of peripheral blood, bone marrow, lymph node, thymus, Peyers patches and spleen.

Minor points:

The term lymphoproliferative disease would suggest lymphocytosis, especially if clonal, lymphadenopathy and infiltration of the bone marrow. I suggest to rethink this term.

Fig. 1f. Please incorporate in the main text the source of CD4+ T cells.

Other than that, I think that the study is excellent.

Reviewer #2 (Remarks to the Author):

Zhu et al in their work entitled 'Microbiota-assisted iron uptake promotes immune tolerance in the intestine' present new evidence of iron/microbiota crosstalk with regards to host Treg functions with their translational impacts on both acute and chronic intestinal inflammation. Intriguingly, contrary to the established notion, where iron (and/or oxygen) deprivation is the main trigger to activate the HIF signaling, their data show that iron availability induces HIF2 expression and function which in turn promotes the transcriptional regulation of Treg functions via c-Maf. A major concern is the paradoxical findings of iron activating HIF-2. Although it has been shown that IRE in HIF2 is functional it is thought to be more of a fine-tuning mechanism. Secondly the authors have not convincingly demonstrated that the phenotype observed is not just a secondary effect of cell death and loss of Treg due to the necessity of iron for most enzymatic functions including mitochondrial respiration and DNA synthesis.

Other concerns:

1. Iron induces c-Maf expression by enhancing IRE-dependent HIF-2a expression: The authors show that HIF2a-5'UTR activity is increased by iron stimulation. To show that the 5'UTR activity is occurring via canonical IRE-IRP interaction, they should address it by showing

- i) Comparison with IRE-mutant 5'UTR-Luc activity, and or
- ii) IRE-IRP interaction, using both WT and mutant IREs

These assays will also address if non-canonical IRE functions are responsible to iron-dependent translational induction of HIF2 expression, which is contrary to the PHD-mediated HIF signaling observed during iron/O₂ deprivation.

Please include your inputs, both in the introduction and discussion sections, in the light of background information with regards to current paradigm of iron-mediated signaling. For example, Tfr_c itself is harboring 5 IREs in its 3'UTR which is its major regulatory pathway in addition to its being well-known HIF target as its promoter contains hypoxia response element (HRE).

2. Lines 188-189, and Suppl. Fig S6f,g: Please include the evidence of efficient HIF2 knockdown.

3. Intracellular iron level analysis:

- i) Calcein-AM based LIP measurement: In the figures (for example, Figures 5A, 5F, S8a and others) it is important to clarify that increased MFI levels mean low intracellular iron to avoid confusion. The description in the figure legend is appropriate. Please also include a statement in the Results section.
- ii) In addition to calcein-AM based labile iron pool (LIP) assessment, the authors need to show more direct evidence of iron level alternations by at least one of the following: a) measurement of iron levels by ICPMS; b) fluorescence study using iron-specific dye such as FerroOrange; c) Ferritin protein levels.

4. A few questions regarding pentanoate:

- i) What is/are its bacterial origin? A discussion including the bacterial source, and their known implication(s) in host homeostasis will improve the readership.
- ii) Is pentanoate exclusively of microbial origin? Is it detectable in GF mice?
- iii) Is it altered in constitutive or inducible Tfr_c KO models discussed in this work?

5. Line 269: " Pentanoate could be an ideal agent" is an overstatement as the authors present the preventive role of the compound in the colitis model. In support of their claim, they need to show the evidence that it attenuates active inflammation when administered after disease onset.

6. Figure 6C: Hematological values show only two samples in each group, which is insufficient evidence for such a crucial conclusion.

7. Also, multiple data have only two samples in one or more groups, which make it difficult to appreciate statistical significance of the evidence presented. For example: Figures 1b, S6b.

Minor concerns:

1. Line 111: Please cite the appropriate Figure for the statement " a magnitude similar to those of the Treg reduction in 6-day-old Tfr_c cKO mice."

2. Lines 189-190: Please add a reference to “Finally, reanalysis of previously published ChIP-Seq results ...”
3. Line 244: Reference # 54 should be replaced by correct reference, which is #69.
4. Line 272: “Expression of FTH ...”, please mention if it will be serum or cellular FTH
5. Line 299: It seems the appropriate phrase would be “iron tug war” between the host and the microbiota ...

Reviewer #3 (Remarks to the Author):

These are interesting and exhaustive studies on the role of iron in immune function and IBD. I have the following issues:

1. In the introduction the statements regarding iron deficiency in IBD require further elaboration. Mechanisms other than bleeding must be involved.
2. The results relating to fecal samples etc. from Crohn's and UC patients are subject to other explanations and in the absence of more direct evidence of a role for pentanoate in UC or Crohn's disease, I would exclude these sections.
3. Though widely used it is important to understand that the DSS model is an injury model and does not necessarily recapitulate the clinical or immunological features of IBD in humans.
4. In the discussion I am sure that the phrase should be "tug of war"

Reviewer #1 (Remarks to the Author):

The current manuscript entitled 'Microbiota-assisted iron uptake promotes immune tolerance in the intestine' by Dr. Zhu and colleagues investigates fundamental biological functions of TfR1 in Tregs using KO models. The study is very comprehensive and well-conducted, and the data are novel and well presented.

The findings are relevant to basic immunology, and the authors demonstrate possible translational aspects in IBD. We appreciate the reviewer's instructive comments on our study and are glad that the reviewer found that our work is "very comprehensive and well-conducted".

I have several major and minor points that I would like to ask the authors to address:

Major points:

Fig. 1a

The authors claim that TfR1 was specifically deleted in Tregs in the *Tfrc* cKO mice. Can the authors provide the isotype control in Fig. 1a? The authors should also demonstrate TfR1 surface levels by flow cytometry on other cell types such as CD4+ T-bet+, CD4+ GATA3+, CD4+ RORgt+ and CD8+ cells in a supplementary figure.

We appreciate the reviewer's suggestion and have added isotype control in the **new Fig. 1a**. Furthermore, we analyzed the TfR1 surface levels in other T cell subsets and found that TfR1 expression was only reduced in Treg cells of *Tfrc* cKO mice. Elevated TfR1 expression in non-Treg cells was likely caused by T cell activation in these mice (**Suppl. Figure S1a and Figure for reviewer 1**).

I realize that some aspects are touched on in FigS1 but I still would like the authors to better characterize the phenotype of these *Tfrc* cKO mice:

What is the reason for the accumulation of CD4+ T cells in tissues? Is it increased proliferation or reduced cell death? To better describe lymphocytic infiltration of organs, the authors should report the absolute numbers of CD4+, CD8+, CD19+ and CD56+ cells in the organs shown by histopathology in Fig. 1d.

Please also include cell numbers and cytology or histopathology of peripheral blood, bone marrow, lymph node, thymus, Peyer's patches, and spleen.

We have conducted a more thorough characterization of the *Tfrc* cKO mice. The accumulation of T cells in tissues was mainly caused by increased cellular proliferation determined by Ki-67 staining (**New Figure S2a and Figure for Reviewer 2a**). In contrast, cell death determined by staining of cleaved caspase-3 was not diminished in the *Tfrc* cKO mice (**New Figure S2b and Figure for Reviewer 2b**).

We also included cell numbers of CD4+, CD8+, CD19+ and CD56+ cells in the liver, lung, blood, bone marrow, spleen, lymph nodes, and thymus (**New Figure S1d**).

Minor points:

The term lymphoproliferative disease would suggest lymphocytosis, especially if clonal, lymphadenopathy and infiltration of the bone marrow. I suggest to rethink this term.

We agree with the reviewer's suggestion and have replaced lymphoproliferative disease with the autoinflammatory disease to more accurately reflect our findings.

Fig. 1f. Please incorporate in the main text the source of CD4+ T cells.

We have added this information to the text.

Other than that, I think that the study is excellent.

Reviewer #2 (Remarks to the Author):

Zhu et al in their work entitled 'Microbiota-assisted iron uptake promotes immune tolerance in the intestine' present new evidence of iron/microbiota crosstalk with regards to host Treg functions with their translational impacts on both acute and chronic intestinal inflammation. Intriguingly, contrary to the established notion, where iron (and/or oxygen) deprivation is the main trigger to activate the HIF signaling, their data show that iron availability induces HIF2 expression and function which in turn promotes the transcriptional regulation of Treg functions via c-Maf. A major concern is the paradoxical findings of iron activating HIF-2. Although it has been shown that IRE in HIF2 is functional it is thought to be more of a fine-tuning mechanism.

We appreciate the expert evaluation of our work and have conducted a thorough analysis to address concerns from the reviewer in our revised manuscript.

We agree with the reviewer that iron deficiencies may enhance HIF-2 α expression by blocking PHD activities which need iron as enzymatic co-factors. Nonetheless, the effects of iron deficiency on HIF-2 α expression may vary depending on the specific cell type and the overall physiological context (*i.e.* oxygen levels). For example, diet-induced iron deficiency inhibited HIF-2 α expression in the liver (PMID: 21753061) but induced HIF-2 α in the intestine (PMID: 19147412).

Thus, we conducted a more thorough analysis of how iron regulates HIF-2 α expression in Treg cells. Partial iron deficiency caused by treatment with a lower dosage of an iron chelator, DFO (2.5 μ M) inhibited HIF-2 α expression in Treg cells (**Figure for Reviewer 3a**). In contrast, more severe iron deficiencies induced by a higher dosage of DFO (20 μ M) promoted HIF-2 α expression (**Figure for Reviewer 3a**). Under the same conditions, expression of HIF-1 α was enhanced by either low or high concentrations of DFO presumably because HIF-1 α does not have IRE elements (**Figure for Reviewer 3b**). These results indicate that iron/IRE-dependent translational control of HIF-2 α expression was more sensitive to iron deprivation than iron/PHD mediated regulation of HIF-2 α protein stability in Treg cells. This is consistent with the notion that regulation at RNA levels is more sensitive than protein stability regulations. Given that *Tfrc*-deficient Treg cells only exhibited a partial reduction of cellular iron, we believe that iron/IRE-mediated translation regulation is the main mechanism regulating HIF-2 α expression downstream of TfR1 in Treg cells.

Secondly, the authors have not convincingly demonstrated that the phenotype observed is not just a secondary effect of cell death and loss of Treg due to the necessity of iron for most enzymatic functions including mitochondrial respiration and DNA synthesis.

We agree with the reviewer that iron deficiencies impair mitochondrial respiration and DNA synthesis, which may directly result in increased cell death and/or reduced cell proliferation. However, we believe that the phenotypes in the *Tfrc* cKO mice were not just caused by the loss of Treg cells based on the following two lines of evidence.

First, substantial amounts of Treg cells exist in *Tfrc* cKO mice before the disease's onset. In 10-day-old *Tfrc* cKO mice, ratios of Treg cells were reduced by around 30%, yet total numbers of Treg in spleen and lymph nodes were comparable to that in wild-type littermates (**Figures for reviewer 4a**). Such discrepancy between Treg numbers and ratios is likely caused by the expansion of conventional (non-Treg) cells in the *Tfrc* cKO mice.

Second, we didn't observe reduced proliferation or increased cell death of Treg cells in *Tfrc* cKO mice determined by Ki-67 staining and cleaved caspase 3 staining respectively (**Figures for reviewers 4b&c**). This is likely because there was only ~50% reduction of cellular iron in TfR1 deficient Treg cells and T cells can uptake non-transferrin associated iron.

Taken together, we believe that the main reason for the lethal autoimmunity in *Tfrc* cKO mice is the loss of Treg effector functions including differentiation toward cMAF⁺ Treg cells.

Figure for reviewer 4. (a). numbers of Treg cells in the spleen and lymph nodes of 10-day old WT or *Tfrc* cKO mice were calculated. (b-c). Proliferation (b) or cell death (c) of Treg cells in 5-day old WT or *Tfrc* cKO mice was determined by staining of Ki67 and cleaved caspase 3. Data are summary (a) or representative (b,c) of three independent experiments

Other concerns:

1. Iron induces c-Maf expression by enhancing IRE-dependent HIF-2 α expression: The authors show that HIF2 α -5'UTR activity is increased by iron stimulation. To show that the 5'UTR activity is occurring via canonical IRE-IRP interaction, they should address it by showing

- i) Comparison with IRE-mutant 5'UTR-Luc activity, and or
- ii) IRE-IRP interaction, using both WT and mutant IREs

These assays will also address if non-canonical IRE functions are responsible to iron-dependent translational induction of HIF2 expression, which is contrary to the PHD-mediated HIF signaling observed during iron/O₂ deprivation.

We appreciate the reviewer's suggestions and have confirmed that IRE is responsible for iron-mediated induction of HIF-2 α . First, we constructed two mutant 5'UTR luciferase reporters of HIF-2 α , both of which abolished interactions between the IRE and IRP1/2. When the reporters were transfected into Treg cells, only the activities of WT reporter, but not that of the mutant reporters, were increased following the iron stimulation (**New Figure S7e, Figure for reviewers 5a**).

Second, previously published studies (PMID: 17417656) have confirmed that IRP-IRE interactions were dependent on IRE as the IRE mutant lost its binding to the IRPs (**Figure for reviewers 5b**, excerpted from PMID: 17417656).

Figure for reviewer 5. IRE element mediates iron-induced HIF2 α expression. (a). iTreg cells were transfected with a HIF-2 α 5'-UTR luciferase reporter or two mutant disrupting the IRE element together with the control Renilla vector by electroporation. Twelve hours after transfection, the cells were stimulated with 50 μ M FeSO₄ for 12 hours, and the luciferase reporter activity were determined with a dual-luciferase kit. The data were normalized to the cells without FeSO₄ stimulation and summarized in three independent experiments. Mutation 1, the 5-CAGUGU-3 loop sequence is changed to 5-CAAAGU-3. mutation2: The IRE loop (CAGUGU) first C was deleted (b) 5'UTR of HIF-2 α bind to IRP1 and IRP2. The unlabeled HIF2 α (EPAS1) 5'UTR RNA competes with the labeled IRE probe from FTH1 for binding to IRP1 (Excerpted from PMID: 17417656).

Please include your inputs, both in the introduction and discussion sections, in the light of background information with regards to current paradigm of iron-mediated signaling. For example, *Tfrc* itself is harboring 5 IREs in its 3'UTR which is its major regulatory pathway in addition to its being well-known HIF target as its promoter contains hypoxia response element (HRE).

We appreciate the reviewer's suggestion and have included a brief introduction on current paradigm of iron-mediated signaling in the introduction (line 44-47) and the discussion (line 334-338)

2. Lines 188-189, and Suppl. Fig S6f,g: Please include the evidence of efficient HIF2 knockdown.

We have included the results showing efficient HIF-2 α knockdown in the shRNA-transduced cells (**Figure for the reviewers 6a**).

3. Intracellular iron level analysis:

i) Calcein-AM based LIP measurement: In the figures (for example, Figures 5A, 5F, S8a and others) it is important to clarify that increased MFI levels mean low intracellular iron to avoid confusion. The description in the figure legend is appropriate. Please also include a statement in the Results section.

We appreciate the reviewer's suggestions and have included a statement that Calcein-AM staining is reversely correlated with levels of intracellular iron in the Results section.

ii) In addition to calcein-AM based labile iron pool (LIP) assessment, the authors need to show more direct evidence of iron level alternations by at least one of the following: a) measurement of iron levels by ICPMS; b) fluorescence study using iron-specific dye such as FerroOrange; c) Ferritin protein levels.

We appreciate the suggestion and have analyzed intracellular Ferritin protein levels in colonic Treg cells. Ferritin expression was much lower in colonic Treg cells from germ-free mice than their counterparts in specific pathogen free mice (SPF) (**Figure for reviewer 6b and new Figure S9a**). Because intracellular Ferritin is correlated with cellular iron levels, these results further strengthen our conclusion that microbiota promotes iron uptake in colonic Treg cells.

4. A few questions regarding pentanoate:

i) What is/are its bacterial origin? A discussion including the bacterial source, and their known implication(s) in host homeostasis will improve the readership.

Previous studies indicate that *Megasphaera* is the main source of pentanoate (PMID 27340431), and we have added this information to the revised manuscript.

ii) Is pentanoate exclusively of microbial origin? Is it detectable in GF mice?

Pentanoate is exclusive of microbial origin because it's undetectable in GF mice (**Figure for reviewers 7a and new Figure S9e**).

iii) Is it altered in constitutive or inducible *Tfrc* KO models discussed in this work?

We found that pentanoate is greatly reduced in the colon of the *Tfrc* cKO mice (**Figure for reviewers 7b**). However, the reduced pentanoate could be caused by either loss of colonic Treg or secondary to the autoinflammatory diseases in these mice.

5. Line 269: “ Pentanoate could be an ideal agent” is an overstatement as the authors present the preventive role of the compound in the colitis model. In support of their claim, they need to show the evidence that it attenuates active inflammation when administered after disease onset.

We appreciate the reviewer’s suggestion and have conducted the experiments with therapeutic settings. In our new experiments, chronic colitis was first induced via repeated administration of DSS, and pentanoate was given to the mice 20 days following the first DSS treatment (**Figures for reviewers 8a**). Similar to what we observed in the previous preventive setting, pentanoate treatment also greatly mitigated both anemia (**Figure for reviewers 8b**) and intestinal inflammation (**Figure for reviewers 8c**) in such a therapeutic setting. These results were included in the new Figure 7.

6. Figure 6C: Hematological values show only two samples in each group, which is insufficient evidence for such a crucial conclusion.

We apologize for the confusion. In the original **Figure 6C**, hematological values were derived from two independent experiments with each group having at least five mice, and the figure was plotted based on the average of each group. In the revised manuscript, we have repeated the experiments with currently 3 independent experiments and at least 15 mice in each group.

7. Also, multiple data have only two samples in one or more groups, which make it difficult to appreciate the statistical significance of the evidence presented. For example: Figures 1b, S6b.

For Figure 1b, we have repeated the experiments with more replicates.

For Figure S6b, the original data were derived from RNA-seq experiments, and we have repeated the experiments using RT-PCR to include more samples.

Minor concerns:

1. Line 111: Please cite the appropriate Figure for the statement “ ... a magnitude similar to those of the Treg reduction in 6-day-old Tfr α cKO mice.”

We have added the citation of the Figure in the revised manuscript.

2. Lines 189-190: Please add a reference to “Finally, reanalysis of previously published ChIP-Seq results ...”

We have added the reference in the revised manuscript.

3. Line 244: Reference # 54 should be replaced by the correct reference, which is #69.

We have corrected the references.

4. Line 272: “Expression of FTH ...”, please mention if it will be serum or cellular FTH

The expression of FTH was based on its cellular mRNA levels, and we have added this information to the manuscript.

5. Line 299: It seems the appropriate phrase would be “iron tug war” between the host and the microbiota ...

We have corrected the phrase to “iron tug war”.

Reviewer #3 (Remarks to the Author):

These are interesting and exhaustive studies on the role of iron in immune function and IBD.

We appreciate the reviewer’s positive comments on our study which is “interesting and exhaustive”.

I have the following issues:

1. In the introduction the statements regarding iron deficiency in IBD require further elaboration. Mechanisms other than bleeding must be involved.

We agree with this reviewer, and have revised the introduction to include more potential mechanisms contributing to iron deficiency in IBD, including inflammation (e.g. Tumor necrosis factor α , PMID: 16566752; IL-6-induced hepcidin expression, PMID: 15124018) and dysbiosis (PMID: 31708445)

2. The results relating to fecal samples etc. from Crohn's and UC patients are subject to other explanations and in the absence of more direct evidence of a role for pentanoate in UC or Crohn's disease, I would exclude these sections.

We understand the concern about the limitations of these results. To further strengthen this conclusion, we analyzed the abundance of *Megasphaera*, known commensal bacteria producing pentanoate, and found they were also reduced in Crohn's and UC patients (Figures for reviewer 9). Although these results are all correlative and subject to alternative explanations, they may provide a promising starting point for future studies. We also included these limitations in the revised manuscript.

3. Though widely used it is important to understand that the DSS model is an injury model and does not necessarily recapitulate the clinical or immunological features of IBD in humans.

We agree with the reviewer and have added to the discussion the limitations of our study.

4. In the discussion I am sure that the phrase should be "tug of war"

We have corrected this typo in the revised manuscript.

REVIEWERS' COMMENTS

Reviewer #1 (Remarks to the Author):

The revised manuscript entitled 'Microbiota-assisted iron uptake promotes immune tolerance in the intestine' by Dr. Zhu and colleagues investigates fundamental biological functions of TfR1 in Tregs using KO models.

The study is very comprehensive and well-conducted, and the data are novel and well presented. The findings are relevant to basic immunology, and the authors demonstrate possible translational and therapeutic aspects in IBD.

I also think that the authors have addressed all concerns originally raised by the reviewers during an extensive revision and that the manuscript is now appropriate for publication in 'Nature Communications'.

Reviewer #2 (Remarks to the Author):

The authors have addressed all major concerns

Reviewer #3 (Remarks to the Author):

No further comments

Reviewer #4 (Remarks to the Author):

This report by Zhu and colleagues investigates the role of iron uptake on the function of Tregs and its consequence on intestinal immune tolerance using KO mouse models. The study is well conducted and, following revision as suggested by three reviewers, the data is clearly presented, well controlled and appropriately discussed. The study is relevant and interesting both from a basic biology perspective and also with applications to IBD in humans.

Minor points:

The comment describing that in infiltrated peripheral organs 'B cells are depleted Tfrc cKO mice' (line 87) should be revised as although this is true in liver, the same is not seen in lung.

Line 98, the term 'mature Treg cells' should be clarified. I assume the authors are referring to Tregs in the periphery rather than in the thymus?

Reviewer #1 (Remarks to the Author):

The revised manuscript entitled 'Microbiota-assisted iron uptake promotes immune tolerance in the intestine' by Dr. Zhu and colleagues investigates fundamental biological functions of TfR1 in Tregs using KO models.

The study is very comprehensive and well-conducted, and the data are novel and well presented. The findings are relevant to basic immunology, and the authors demonstrate possible translational and therapeutic aspects in IBD.

I also think that the authors have addressed all concerns originally raised by the reviewers during an extensive revision and that the manuscript is now appropriate for publication in 'Nature Communications'.

Reviewer #2 (Remarks to the Author):

The authors have addressed all major concerns

Reviewer #3 (Remarks to the Author):

No further comments

Reviewer #4 (Remarks to the Author):

This report by Zhu and colleagues investigates the role of iron uptake on the function of Tregs and its consequence on intestinal immune tolerance using KO mouse models. The study is well conducted and, following revision as suggested by three reviewers, the data is clearly presented, well controlled and appropriately discussed. The study is relevant and interesting both from a basic biology perspective and also with applications to IBD in humans.

Minor points:

The comment describing that in infiltrated peripheral organs 'B cells are depleted Tfrc cKO mice' (line 87) should be revised as although this is true in liver, the same is not seen in lung.

We appreciate the reviewer's suggestions and have revised the manuscript to state that 'B cells were depleted in the spleen, PBMC, bone marrow, and liver'. We believe this updated phrasing more accurately conveys the results of our study.

Line 98, the term 'mature Treg cells' should be clarified. I assume the authors are referring to Tregs in the periphery rather than in the thymus?

We have clarified the term "mature Treg cells" and changed it to 'Treg cells in the periphery'.